# Wasserstein Geometry-Aware Adaptive Control via Meta-Learning

**Xingyu Yang** [1]   **Hanzhang Qu** [1]   **Ye Cao** [1]   **Jianfu Cao** [1]

## Abstract

Adaptive control of nonlinear systems under unknown disturbances requires learning algorithms aligned with the downstream control objective. While control-oriented meta-learning addresses the mismatch between regression-based identification and tracking performance, existing methods rely on Euclidean or static algebraic geometries that fail to capture the distributional structure of system uncertainties. We propose a framework that lifts adaptation into Wasserstein space, measuring parameter estimation errors as the optimal transport cost between estimated and true system behaviors. By constructing a Wasserstein Bregman divergence over representative task distributions, we use meta-learning to jointly optimize nonlinear feature representations, control gains, and transport geometry. This adaptation law learns an adaptation geometry that captures structural properties of the underlying physical system, implementing a physically grounded, data-driven attention mechanism. Closed-loop tracking simulations demonstrate that our controller achieves optimal performance on both fully-actuated and underactuated nonlinear planar rotorcraft, maintaining robustness under significant distributional shifts between training and testing conditions.

## 1. Introduction

High-performance robotic control is challenged by nonlinear dynamics and complex environment interactions. While physics-based models combined with nonlinear control laws can guarantee desirable theoretical properties, such as asymptotic tracking, they rely on the assumption that uncertainties enter the dynamics in a known and structured form. In realistic settings, specifying the structure of all possible disturbances a priori is often infeasible. This limitation necessitates adaptive control approaches that can adjust controller parameters online to compensate for unknown variations.

Although recent advances in deep learning provide expressive function approximators for dynamical systems, deploying such rich models in fast feedback loops is often impractical due to computational latency. More fundamentally, standard machine learning approaches suffer from an objective mismatch: they are primarily regression-oriented, minimizing prediction error under the assumption that improved model accuracy directly translates to improved control performance. In contrast, decades of adaptive control theory demonstrate that learning should be aligned with the downstream control task—closed-loop tracking guarantees can be achieved without precise parameter identification, as long as the learning process is explicitly control-oriented (Gevers, 2005; Ljung, 1999).

Recent progress in control-oriented meta-learning has addressed this objective mismatch by directly optimizing adaptive controllers with closed-loop tracking error as the meta-objective (Richards et al., 2021; 2023). By meta-learning nonlinear feature representations and control gains from data, these methods successfully bridge the gap between data-driven learning and classical adaptive control theory. Building on this foundation, Tang et al. (2025) introduced Mirror Descent into the adaptation law, enabling non-Euclidean update geometries through learnable Bregman divergences parameterized by $p$-norms. However, these approaches remain confined to static, algebraic geometries that do not explicitly account for the distribution of system states encountered during closed-loop operation.

In this paper, we propose a framework that lifts adaptive control from finite-dimensional parameter spaces to the space of probability distributions equipped with Wasserstein geometry. Motivated by the insight that parameter estimation errors should reflect the optimal transport cost between estimated and true system behaviors, we construct a Wasserstein Bregman divergence over representative task distributions, which we implement via a tractable particle-based variational approach. We leverage meta-learning to jointly optimize nonlinear feature representations, control gains, and the underlying transport geometry, enabling the con-

[1]State Key Laboratory for Manufacturing Systems Engineering and the School of Electronic and Information Engineering, Xi'an Jiaotong University, Xi'an 710049, China. Correspondence to: Jianfu Cao <cjf@mail.xjtu.edu.cn>.

*Proceedings of the $43^{rd}$ International Conference on Machine Learning*, Seoul, South Korea. PMLR 306, 2026. Copyright 2026 by the author(s).

troller to discover adaptation laws that implicitly prioritize updates in frequently visited state-space regions. Crucially, we extend this geometry-aware framework to underactuated nonlinear systems, addressing a setting overlooked by prior Mirror Descent-based approaches. Extensive empirical validation demonstrates that our method significantly outperforms other baselines, achieving superior tracking performance and robustness under significant distributional shifts between training and testing conditions.

## 2. Related Work

In this section, we situate our work within three key areas: adaptive control, mirror descent and Wasserstein geometry, and meta-learning for control.

### 2.1. Adaptive Control

Classical adaptive control theory shows that closed-loop stability can be achieved even when parameter estimates do not converge to their true values, provided the adaptation law prioritizes the control objective over parameter identification accuracy (Åström & Wittenmark, 1995; Sastry & Bodson, 1989; Slotine & Li, 1991). This control-oriented perspective was formalized for linear systems through iterative closed-loop experiment design (Gevers, 1993; Hjalmarsson et al., 1996), and extended to nonlinear systems via Lyapunov-based adaptation laws that guarantee tracking convergence under structured uncertainties (Krstic et al., 1995). Recent work has incorporated neural networks as function approximators within the adaptive loop, either for learning Lyapunov functions jointly with controllers (Chang et al., 2019; Dai et al., 2021), or for representing unknown dynamics features updated online (Sanner & Slotine, 1992; Joshi et al., 2021). Our work builds on this foundation by extending the adaptation law from Euclidean to Wasserstein geometry, enabling the controller to exploit the distributional structure of system uncertainties.

### 2.2. Mirror Descent and Wasserstein Geometry

Mirror descent generalizes gradient descent to non-Euclidean spaces by adapting updates to the geometry of the feasible set via a Bregman divergence (Nemirovskij & Yudin, 1983; Beck & Teboulle, 2003). In reinforcement learning and control, mirror descent has been employed to ensure stable policy updates by constraining the KL-divergence between policies (Schulman et al., 2015; Tomar et al., 2022). Tang et al. (2025) recently introduced mirror descent into adaptive control, meta-learning algebraic potential functions (e.g., $p$-norms) to improve convergence rates. However, these approaches remain confined to static, finite-dimensional vector spaces. Optimal transport provides a framework for measuring distances between probability distributions, with the Wasserstein metric offering continuity

under distributional shift and informative gradients even for distributions with disjoint supports (Villani, 2008; Peyré & Cuturi, 2019). These properties have driven its adoption in machine learning, notably in Wasserstein GANs, where transport-based losses avoid the vanishing gradient issues of KL-based objectives (Arjovsky et al., 2017). Bonet et al. (2024) established theoretical foundations for mirror descent in Wasserstein space, providing convergence guarantees for optimization over probability measures. While these advances have primarily targeted sampling and generative modeling, we adapt the Wasserstein geometry framework to adaptive control, constructing a Bregman divergence that measures parameter estimation errors as optimal transport costs. The details can be found in Appendix A.

### 2.3. Meta-Learning for Control

A common approach to learning-based control is to meta-learn a dynamics model offline that can be quickly adapted to new tasks using limited online data. Bertinetto et al. (2019) and Lee et al. (2019) backpropagate through closed-form ridge regression solutions for few-shot learning with a maximum likelihood meta-objective. O'Connell et al. (2022) apply this method to learn neural network features for mechanical systems, while Harrison et al. (2018; 2020) backpropagate through Bayesian regression to train prior dynamics models with nonlinear features. Clavera et al. (2019) use gradient descent on a multi-step likelihood objective as the base-learner. These regression-oriented methods minimize prediction error under the assumption that improved model accuracy translates directly to better control performance. However, this assumption often fails: the objective mismatch between regression accuracy and closed-loop tracking can yield suboptimal controllers (Recht, 2019). Richards et al. (2021; 2023) addressed this gap by proposing control-oriented meta-learning, which directly optimizes closed-loop tracking error as the meta-objective while learning nonlinear feature representations and control gains. While effective, their approach restricts the meta-learner to optimizing feature representations within a fixed Euclidean geometry. Our work generalizes this line of research by meta-learning the full transport geometry in Wasserstein space, rather than restricting to finite-dimensional parameter spaces or scalar exponents.

## 3. Problem Statement

Consider the uncertain nonlinear system with unknown disturbance,

$$\dot{x} = f(x) + g(x)(u + f_{\text{ext}}(x, w)), \quad (1)$$

where $x \in \mathbb{R}^n$ denotes the state vector, $w \in \mathbb{R}^{d_w}$ is some unknown disturbance, $u \in \mathbb{R}^m$ denotes the control input, and $f(x) : \mathbb{R}^n \to \mathbb{R}^n$ and $g(x) : \mathbb{R}^m \to \mathbb{R}^n$ are known

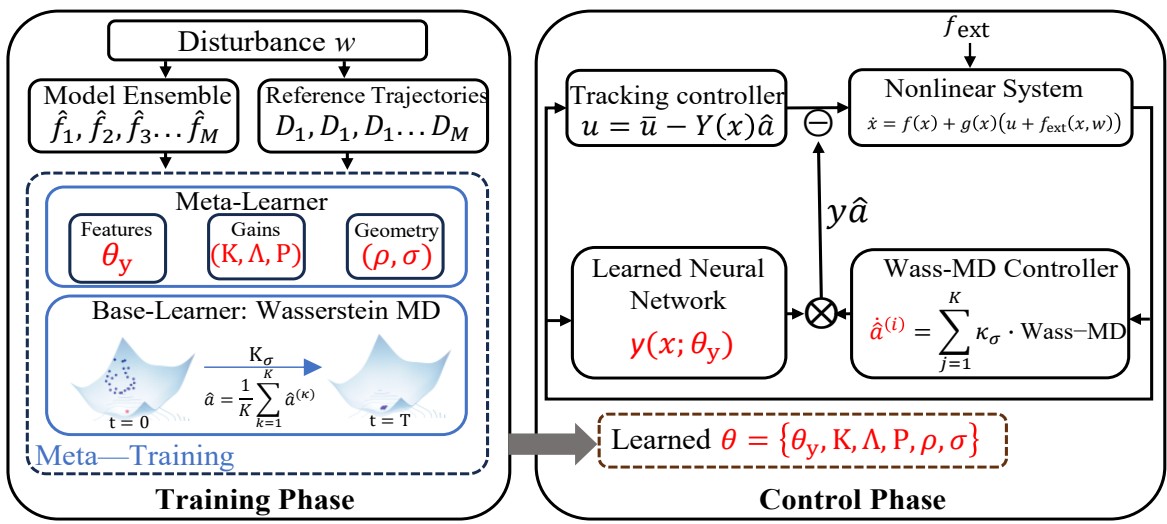

*Figure 1.* An illustration of our proposed method. Offline-learned parameters used for online control are highlighted in red

smooth nonlinear functions. And $f_{\text{ext}}(x, w) : \mathbb{R}^n \times \mathbb{R}^{d_w} \to \mathbb{R}^m$ is some unknown external disturbance. Generally, for a given reference trajectory $r(t) \in \mathbb{R}^n$, we want to design controller $u(t)$ such that $x(t)$ converges to $r(t)$, i.e., system (1) can track the reference trajectory $r(t)$.

**Assumption 3.1.** The disturbance can be linearly parameterized as $f_{\text{ext}}(x, w) = Y(x)a$, where $Y : \mathbb{R}^n \to \mathbb{R}^{m \times d}$ is a known feature function and $a \in \mathbb{R}^d$ is a vector of unknown parameters.

If disturbance $f_{\text{ext}}(x, w)$ does not exist ($f_{\text{ext}}(x, w) = 0$), a nominal controller $\bar{u}(t)$ can be easily designed to achieve asymptotic tracking, i.e., $x(t) \to r(t)$ as $t \to \infty$. Because of the existence of $f_{\text{ext}}$, we want to design an adaptive law to cancel the disturbance's effect. According to Assumption 3.1, a reasonable feedback law can be designed as follows,

$$u = \bar{u} - Y\hat{a}. \tag{2}$$

The adaptation law $\dot{\hat{a}} = \rho(x, r, \hat{a})$ makes the state $x(t)$ track $r(t)$, a goal that is distinct from requiring $\hat{a}$ to converge to the true parameters $a$. Since $Y(x)a$ depends on $x(t)$ and hence indirectly on the target $r(t)$, the adaptation process is inextricably linked to the feedback loop and the reference trajectory. Consequently, adaptive control estimates uncertainty to cancel $Y(x)a$ for closed-loop stability, prioritizing tracking performance over open-loop parameter recognition.

## 4. Bi-Level Meta-Learning

We briefly review the fundamentals of meta-learning (Finn et al., 2017) to establish the basis for our adaptive control framework. In conventional machine learning, the objective is typically to find optimal parameters $\psi^*$ that minimize a loss function $\mathcal{L}$ over a single dataset $\mathcal{D}$, i.e.,

$\psi^* \in \arg\min_\psi \mathcal{L}(\psi, \mathcal{D})$. In contrast, meta-learning considers a distribution of tasks. For a collection of $M$ tasks, each task $i$ is associated with a specific loss function $\mathcal{L}_i$, a training dataset $\mathcal{D}_i^{\text{train}}$, and an evaluation dataset $\mathcal{D}_i^{\text{eval}}$. A key component is the adaptation mechanism, denoted as Adapt $: (\theta, \mathcal{D}_i^{\text{train}}) \to \psi_i$, which maps the shared meta-parameters $\theta$ and task-specific training data to task-specific parameters $\psi_i$. Consequently, meta-learning is formalized as the following bi-level optimization problem:

$$\theta^* \in \arg\min_\theta \frac{1}{M} \left( \sum_{i=1}^M \mathcal{L}_i(\psi_i, \mathcal{D}_i^{\text{eval}}) + \mu_{\text{meta}} \|\theta\|_2^2 \right) \tag{3}$$
$$\text{s.t. } \psi_i = \text{Adapt}(\theta, \mathcal{D}_i^{\text{train}})$$

where $\mu_{\text{meta}} \geq 0$ denotes a regularization coefficient. The resulting optimal meta-parameters $\theta^*$ are optimized to enable rapid adaptation across the task distribution, embodying the paradigm of "learning to learn". We refer to the optimization problem in (3) as the meta-problem, and its objective as the meta-loss. In this bi-level formulation, the inner adaptation operator Adapt acts as the base-learner, while the outer-loop optimization procedure that solves (3) serves as the meta-learner.

## 5. Proposed Method: Adaptive Control with Meta-Learned Wasserstein Bregman Divergence

While recent control-oriented meta-learning approaches (Richards et al., 2021; O'Connell et al., 2022) have advanced tracking control, they predominantly rely on Euclidean gradient descent, often failing to capture the data-dependent structure of uncertainties. We transcend these Euclidean and static Mirror Descent limitations by lifting

adaptation into the Wasserstein space. Instead of relying on pre-defined algebraic potentials, we leverage meta-learning to automate the construction of a Wasserstein Bregman divergence induced by the system's occupancy measure $\mu$. This formulation minimizes the physical transport cost of parameter errors rather than a static vector distance, significantly enhancing tracking performance in heterogeneous environments.

We formulate this as a multi-task learning problem where the meta-learner captures task-agnostic structural knowledge to facilitate rapid adaptation. As detailed in Section 5.1 and 5.2, the framework couples a Wasserstein Mirror-Descent base-learner, which executes geometry-aware adaptation, with a meta-learner that needs to learn a suitable Wasserstein Mirror-Descent potential function and the feature basis. In particular, we construct a neural network to learn unknown features $f_{\text{ext}}(x, w)$ from data, denoted as $\hat{f}_{\text{ext}}(x, w) = y(x; \theta_y)a$ where $\theta_y$ are the weights of the network. Furthermore, to enable offline meta-training without access to the true system dynamics, we introduce Model Ensembles in Section 5.3. These ensembles approximate the distribution of tasks from historical data, providing a robust generative environment for learning the optimal control geometry. Figure 1 illustrates the details of our method.

## 5.1. Meta-Learning for Nonlinear Systems

To formulate the meta-learning problem, we define the training data associated with each task as a pair consisting of a reference trajectory $r_i(t) \in \mathbb{R}^n$ and a disturbance signal $w_j(t) \in \mathbb{R}^{d_w}$ over a time horizon $T > 0$. These data are used to optimize the static parameters $\theta := (\theta_\pi, \theta_\rho)$ of the adaptive controller:

$$
\begin{aligned}
u &= \pi(x, r, \hat{a}; \theta_\pi), \\
\dot{\hat{a}} &= \rho(x, r, \hat{a}; \theta_\rho),
\end{aligned}
\tag{4}
$$

such that the pair $(\pi, \rho)$ ensures good tracking of $r_i(t)$ for $t \in [0, T]$ subject to the disturbance $w_j(t)$. In this framework, the adaptation mechanism corresponds to the continuous-time forward simulation of the closed-loop system. Consequently, the task-specific variable $\psi_{ij}$ in (3) is defined as the resulting closed-loop trajectory $\psi_{ij} = \{x_{ij}(t), \hat{a}_{ij}(t), u_{ij}(t)\}_{t \in [0, T]}$, where

$$
\begin{aligned}
x_{ij}(t) &= x_{ij}(0) + \int_0^t f\big(x_{ij}(\epsilon), u_{ij}(\epsilon), w_j(\epsilon)\big) \, d\epsilon, \\
\hat{a}_{ij}(t) &= \hat{a}_{ij}(0) + \int_0^t \rho\big(x_{ij}(\epsilon), u_{ij}(\epsilon), w_j(\epsilon); \theta_\rho\big) \, d\epsilon, \\
u_{ij}(t) &= \pi\big(x_{ij}(t), r_i(t), \hat{a}_{ij}(t); \theta_\pi\big).
\end{aligned}
\tag{5}
$$

This trajectory is computed via numerical integration using standard ODE solvers. For initialization, we set the system

state to match the reference, $x_{ij}(0) = r_i(0)$, and the adaptive parameters to zero, $\hat{a}_{ij}(0) = 0$. The associated task loss is defined as the time-averaged tracking error for the given reference–disturbance pair:

$$
\mathcal{L}_{ij}(\psi_{ij}, D_{ij}^{\text{eval}}) = \frac{1}{T} \int_0^T \Big( \|x_{ij}(t) - r_i(t)\|_2^2 + \alpha \|u_{ij}(t)\|_2^2 \Big) \, dt,
\tag{6}
$$

where $\alpha \geq 0$ regularizes the control magnitude. This LQR-style loss can be readily extended to general weighted norms. By considering $N$ reference trajectories $\{r_i(t)\}_{i=1}^N$ and $M$ disturbance profiles $\{w_j(t)\}_{j=1}^M$, we generate $NM$ training tasks. Integrating the system constraints (5)–(6) into the meta-learning objective (3), we arrive at the following optimization problem:

$$
\begin{aligned}
\min_\theta \quad & \frac{1}{NMT} \Big( \sum_{i=1}^N \sum_{j=1}^M \int_0^T c_{ij}(t) \, dt + \mu_{\text{meta}} \|\theta\|_2^2 \Big) \\
\text{s.t.} \quad & c_{ij} = \|x_{ij} - r_i\|_2^2 + \alpha \|u_{ij}\|_2^2, \\
& \dot{x}_{ij} = f(x_{ij}, u_{ij}, w_j), \quad x_{ij}(0) = r_i(0), \\
& u_{ij} = \pi(x_{ij}, r_i, \hat{a}_{ij}; \theta_\pi), \\
& \dot{\hat{a}}_{ij} = \rho(x_{ij}, r_i, \hat{a}_{ij}; \theta_\rho), \quad \hat{a}_{ij}(0) = 0.
\end{aligned}
\tag{7}
$$

Minimizing the objective in (7) yields the optimal meta-parameters $\theta = (\theta_\pi, \theta_\rho)$ that ensure robust closed-loop tracking, on average, across the specified distribution of reference trajectories $\{r_i\}_{i=1}^N$ and disturbances $\{w_j\}_{j=1}^M$. In practice, the adaptation dynamics are implemented via an interacting particle approximation of the corresponding Wasserstein gradient flow. In our framework, these meta-parameters decompose into three components: the neural network weights $\theta_y = \{W^{(\ell)}, b^{(\ell)}\}_{\ell=1}^L$ that parameterize the nonlinear feature function $y(x; \theta_y)$ for disturbance compensation, the feedback and adaptation gains $(K, \Lambda, P)$ that govern the nominal tracking and learning dynamics, and the transport geometry parameters $(p, \sigma)$ that define the Wasserstein Bregman divergence. By jointly optimizing all three components through (7), our framework provides greater flexibility than prior work that learns only features (**?**) or only the potential exponent (Tang et al., 2025).

To optimize the meta-parameters $\theta$, we employ gradient-based methods, necessitating back-propagation through the continuous-time dynamics, which is efficiently implemented via the adjoint sensitivity method following forward integration (Chen et al., 2018). For our Wasserstein Mirror Descent strategy, these trajectories generate representative task distributions that define the underlying transport geometry. Consequently, by incorporating reference trajectories into the training set, we not only minimize tracking error but also actively shape the adaptation geometry to align the offline meta-learning process with the specific downstream control requirements.

## 5.2. Base-Learner: Wasserstein Geometry-Aware Adaptive Control

We now present the core methodology of this paper, leveraging the meta-learning concepts from Section 4 to address the control problem in (1). To facilitate the exposition, we initially assume that the system dynamics $f$ can be simulated offline and that a dataset of $M$ sample disturbance trajectories $\{w_j(t)\}_{j=1}^M$ is available over $t \in [0, T]$. These assumptions will be relaxed in Section 5.3.

Standard adaptive control theory (Slotine & Li, 1991) offers a baseline for tracking in such systems. Given a reference $r(t)$, a nominal control law $\bar{u}(t)$ exists that achieves asymptotic tracking in the absence of external disturbances ($f_{\text{ext}} = 0$). This nominal performance is grounded in a Lyapunov function $\bar{V}(x, r)$, which ensures that the tracking error converges to zero asymptotically in the nominal closed-loop dynamics. Under the assumption that the external disturbance is linearly parameterizable as $f_{\text{ext}}(x, w) = Y(x)a$, a classical adaptive control strategy can be derived. We lift the adaptation law from finite-dimensional parameter spaces to Wasserstein space. Instead of maintaining a single parameter estimate $\hat{a} \in \mathbb{R}^d$, we represent the parameter uncertainty as a probability measure $\nu_{\hat{a}}$ over $\mathbb{R}^d$ and perform adaptation in the space of probability measures equipped with Wasserstein geometry. For a base potential $\chi : \mathbb{R}^d \to \mathbb{R}$, we define the potential functional over probability measures as:

$$\Psi_\mu(\nu) := \int_{\mathbb{R}^d} \chi(a) \, d\nu(a), \tag{8}$$

where the subscript $\mu$ indicates that the geometry is induced by a reference measure $\mu$. The Wasserstein Bregman divergence between two probability measures $\nu$ and $\nu'$ is then defined as:

$$d_{\Psi_\mu}(\nu \| \nu') := \Psi_\mu(\nu) - \Psi_\mu(\nu') - \langle \nabla_\mu \Psi_\mu(\nu'), \nu - \nu' \rangle, \tag{9}$$

where $\nabla_\mu \Psi_\mu$ denotes the first variation (Wasserstein gradient) of $\Psi_\mu$. The corresponding Lyapunov-like function becomes:

$$V(x, r, \nu_a, \nu_{\hat{a}}) = \bar{V}(x, r) + d_{\Psi_\mu}(P_\# \nu_a \| P_\# \nu_{\hat{a}}), \tag{10}$$

where $\nu_a = \delta_a$ is the Dirac measure at the true parameter $a$, $\nu_{\hat{a}}$ is the distribution of parameter estimates, and $P_\#$ denotes the pushforward by the linear map $a \mapsto Pa$ with $P = P^\top \succ 0$.

A key distinction of our approach is that the reference measure $\mu$ is not arbitrarily chosen, but is induced by the distribution of states visited during closed-loop operation. For a trajectory $x(t)$ over $[0, T]$, this measure is $\mu_T := \frac{1}{T} \int_0^T \delta_{x(t)} \, dt$, where $\delta_{x(t)}$ denotes the Dirac measure at $x(t)$. In the meta-learning setting with $N$ reference

trajectories and $M$ disturbance profiles, the aggregate occupancy measure becomes:

$$\mu := \frac{1}{NM} \sum_{i=1}^N \sum_{j=1}^M \mu_T^{(i,j)}, \tag{11}$$

where $\mu_T^{(i,j)}$ is the occupancy measure of the closed-loop trajectory under reference $r_i$ and disturbance $w_j$. In practice, we approximate the parameter distribution $\nu_{\hat{a}}$ using an empirical measure supported on $K$ particles:

$$\nu_{\hat{a}} \approx \hat{\nu}_K := \frac{1}{K} \sum_{k=1}^K \delta_{\hat{a}^{(k)}}. \tag{12}$$

Following Bonet et al. (2024), we adopt the $p$-norm potential $\chi(a) = \|a\|_p^p = \sum_{i=1}^d |a_i|^p$ with $p > 1$, whose gradient and Hessian are:

$$\begin{aligned} \nabla \chi(a) &= p \cdot \text{sign}(a) \odot |a|^{p-1}, \\ \nabla^2 \chi(a) &= p(p-1) \cdot \text{diag}(|a_i|^{p-2}). \end{aligned} \tag{13}$$

The potential functional over the empirical measure becomes $\Psi_\mu(\hat{\nu}_K) = \frac{1}{K} \sum_{k=1}^K \chi(\hat{a}^{(k)})$. To simplify notation, we define the mirror descent direction:

$$\mathcal{H}(\hat{a}) \triangleq -P^{-1}(\nabla^2 \chi(P\hat{a}))^{-1} P^{-1} Y(x)^\top g(x)^\top \nabla_x \bar{V}(x, r). \tag{14}$$

The Wasserstein gradient flow on this particle system yields the following coupled dynamics for the $i$-th particle:

$$\dot{\hat{a}}^{(i)} = \sum_{j=1}^K \kappa_\sigma(\hat{a}^{(i)}, \hat{a}^{(j)}) \cdot \frac{1}{K} \cdot \mathcal{H}(\hat{a}^{(j)}), \tag{15}$$

where $\kappa_\sigma(a, b) = \exp(-\|a - b\|^2 / 2\sigma^2)$ is a Gaussian kernel with bandwidth $\sigma > 0$. The kernel smoothing approximates the Wasserstein gradient by enabling interaction among particles: each particle's update is influenced by nearby particles, with the bandwidth $\sigma$ controlling the locality of this interaction. Crucially, the kernel interaction term $\kappa_\sigma$ couples the particles, strictly distinguishing our method from a simple ensemble of $K$ independent Mirror Descent controllers (Tang et al., 2025) (which corresponds to the limit $\sigma \to 0$). This coupling forces the particle system to evolve cooperatively along the Wasserstein gradient flow, minimizing the transport cost, whereas an independent ensemble would merely minimize the average Euclidean error without geometric awareness. The final parameter estimate used in the controller is the empirical mean:

$$\hat{a} = \frac{1}{K} \sum_{k=1}^K \hat{a}^{(k)}. \tag{16}$$

Functional analysis shows that the Lyapunov function (10) serves as a valid stability certificate for tracking error ultimately uniformly bounded under the proposed particle-based adaptation law; we refer the reader to Appendix B for the detailed proof.

This formulation yields two important properties. First, the kernel bandwidth $\sigma$ and particle distribution jointly encode the local geometry of the parameter space; by meta-learning $\sigma$ alongside the feature network $\theta_y$ and potential exponent $p$, we obtain a transport geometry tailored to the structure of the underlying dynamical system through meta-learning. Second, the kernel weighting in (15) assigns higher influence to particles in dense regions, prioritizes updates in frequently visited parameter-space regions. By meta-learning the transport geometry alongside the feature network, our method achieves improved robustness compared to static geometries, as demonstrated in our distributional shift experiments.

### 5.3. Model Ensembles

Computing the task loss $\mathcal{L}_j$ in the bi-level problem (7) requires forward integration of system dynamics that depend on the unknown disturbance $f_{\text{ext}}(x; w)$. As the disturbance model is unavailable during offline meta-training, we adopt a data-driven approach inspired by model-based reinforcement learning (Clavera et al., 2018). We construct an ensemble of neural networks $\{\hat{f}_j\}_{j=1}^{M}$ as surrogate dynamics models, each trained on state–action trajectories collected under a fixed disturbance condition $w_j$. The ensemble captures local disturbance-dependent dynamics and replaces the ground-truth system in the inner loop, enabling differentiable trajectory optimization. Training details and the resulting meta-learning objective are provided in Appendix C.

## 6. Experiments

We conduct numerical simulations of our proposed method on two example systems: a Planar Fully-Actuated Rotorcraft (PFAR), and the classic underactuated Planar Vertical Take-Off and Landing (PVTOL) vehicle from Hauser et al. (1992), following the equations of motion for the planar rotorcraft with the specifications below:

$$\begin{pmatrix} \ddot{x} \\ \ddot{y} \\ \ddot{\phi} \end{pmatrix} = \underbrace{\begin{pmatrix} 0 \\ -g \\ 0 \end{pmatrix}}_{f_g} + \underbrace{\begin{bmatrix} \cos\phi & -\sin\phi & 0 \\ \sin\phi & \cos\phi & 0 \\ 0 & 0 & 1 \end{bmatrix}}_{R(\phi)} u + f_{\text{ext}}, \quad (17)$$

while the PVTOL system dynamics are given by

$$\begin{pmatrix} \ddot{x} \\ \ddot{y} \\ \ddot{\phi} \end{pmatrix} = \underbrace{\begin{pmatrix} 0 \\ -g \\ 0 \end{pmatrix}}_{f_g} + R(\phi) \underbrace{\begin{bmatrix} 0 & 0 \\ 1 & 0 \\ 0 & 1 \end{bmatrix}}_{B(\phi)} u + f_{\text{ext}}, \quad (18)$$

where $g = 9.81$ m/s$^2$ is the gravitational acceleration, and the system states $q = (x, y, \phi)$ include the planar quadrotor's center of mass position $(x, y)$ and its roll angle $\phi$ along with their time derivatives $(\dot{x}, \dot{y}, \dot{\phi})$. $R(\phi)$ is a rotation matrix between inertial and body-fixed frames.

The PFAR system is fully-actuated with control input $u \in \mathbb{R}^3$ that directly commands the thrust along the body-fixed $x$-axis, thrust along the body-fixed $y$-axis, and torque about the center of mass. On the other hand, the PVTOL system is underactuated with control input $u \in \mathbb{R}^2$ that directly commands only the thrust along the body $x$-axis and torque about the center of mass.

The unknown disturbance $f_{\text{ext}}(q, \dot{q}; w)$ is modeled as the drag force caused by a wind blowing along the inertial $x$-axis at a speed $w \in \mathbb{R}$, which is defined by the following equations:

$$\begin{aligned} v_x &= (\dot{x} - w)\cos\phi + \dot{y}\sin\phi, \\ v_y &= -(\dot{x} - w)\sin\phi + \dot{y}\cos\phi, \\ v_L &= -\dot{\phi} - v_y, \\ v_R &= \dot{\phi} - v_y, \end{aligned} \quad (19)$$

$$f_{\text{ext}}(q, \dot{q}; w) = -R(\phi)\begin{pmatrix} \beta_1 v_x|v_x| \\ \beta_2 v_y|v_y| \\ \beta_3(v_R|v_R| - v_L|v_L|), \end{pmatrix}$$

where $\beta_1, \beta_2, \beta_3 > 0$ are mass-normalized drag coefficients. Each body-fixed component of the drag force acts in opposition to the corresponding body-fixed component of the vehicle's linear or angular velocity. For the PVTOL system, the disturbance term $f_{\text{ext}}$ constitutes a matched uncertainty if and only if $\beta_1 = 0$. To instead induce an unmatched uncertainty—and thereby create a more challenging disturbance for the PVTOL system—we select a small but nonzero value of $\beta_1$. Specifically, we set $\beta_1 = 0.01$, $\beta_2 = 1$, and $\beta_3 = 1$ in all experiments. These robotic platforms are modeled as Lagrangian dynamical systems; the specific control designs and their stability analyses are detailed in Appendix D.

### 6.1. Experimental Setup

**Training Data.** For both systems, we generate $M = 500$ closed-loop trajectories using a baseline PID controller with fixed gains. Each trajectory is simulated over a time horizon of $T = 30$ seconds with a time step of $dt = 0.01$ seconds. Reference trajectories are generated as smooth random splines in the flat output space $(x, y)$ with 6 knots and polynomial order 9, constrained to have bounded step sizes in the range $[-2, 2]$ meters. Wind speeds $w$ are sampled from a scaled Beta distribution, $w = w_{\min} + (w_{\max} - w_{\min}) \cdot \text{Beta}(5, 9)$, with $w_{\min} = 0$ m/s and $w_{\max} = 6$ m/s, which concentrates samples at moderate wind speeds while maintaining coverage of the training range. This distribution is visualized as $p_{\text{train}}(w)$ in Figure 2.

**Model Ensemble Training.** From the collected trajectories, we train an ensemble of $M$ neural network models, each with 2 hidden layers of 32 units and $\tanh$ activation. Each model $\hat{f}_j$ is trained on a single trajectory $\mathcal{T}_j$ using the

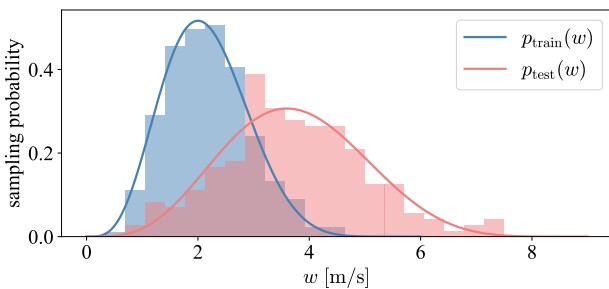

Figure 2. Wind distributions

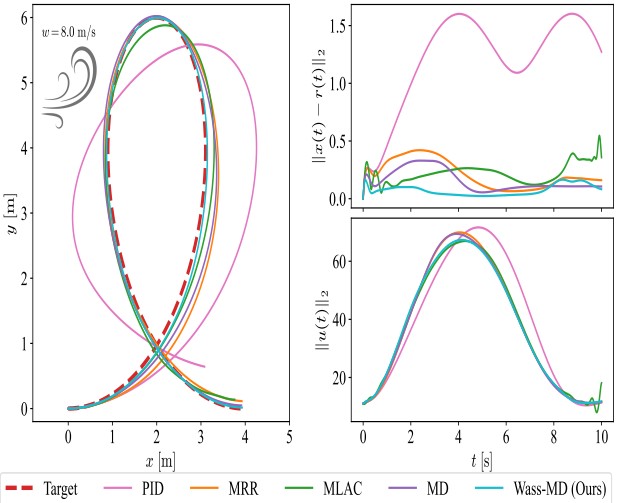

Figure 3. Tracking results for the PFAR on a test trajectory with w = 8.0 m/s

one-step prediction loss with $L_2$ regularization coefficient $\mu_{\text{ensem}} = 10^{-4}$. Training proceeds for 1000 epochs using the Adam optimizer with learning rate $10^{-2}$.

**Meta-Training.** The meta-learner optimizes the feature network parameters $\theta_y$ (2 hidden layers of 32 units), control gains, adaptation gains, the potential exponent $p$, and kernel bandwidth $\sigma$ over $N = 10$ reference trajectories and the full model ensemble. The meta-loss (7) is minimized using Adam with learning rate $10^{-2}$ for 500 steps, with $L_2$ regularization $\mu_{\text{meta}} = 10^{-4}$ and control regularization $\alpha = 10^{-3}$. We use a particle system ($K = 5$) to represent the parameter distribution in Wasserstein space with initial kernel bandwidth $\sigma = 0.1$.

**Baselines.** We compare our meta-trained Wasserstein geometry-aware adaptive controller (Wass-MD) against four baselines: (1) PID control with feed-forward compensation, (2) Meta-Ridge Regression (MRR) (Bertinetto et al., 2019) that meta-learns parametric features via closed-form ridge regression, (3) Meta-Learning Adaptive Control (MLAC) (?) with gradient-based adaptation, and (4) Meta-trained Mirror Descent (MD) (Tang et al., 2025) with learned $p$-norm potential. Detailed formulations of all baselines are provided in Appendix E.

**Testing with Distribution Shift.** To evaluate robustness under distributional shift, we test all controllers on wind conditions sampled from a shifted Beta distribution with $w_{\min} = 2$ m/s and $w_{\max} = 8$ m/s, denoted $p_{\text{test}}(w)$ in Figure 2. This shift increases both the minimum and maximum wind speeds beyond the training range, testing the controllers' ability to generalize to out-of-distribution disturbances.

### 6.2. Results and Discussion

**PFAR: Fully-Actuated System.** Figure 3 shows the tracking performance of all controllers on a loop trajectory under wind speed $w = 8.0$ m/s—beyond the training distribution. All meta-learned controllers successfully complete the loop trajectory, while the PID controller exhibits significant de-

viation from the reference path. In the position–velocity error plot (upper right), Wass-MD achieves the fastest error reduction and maintains the smallest steady-state error. The control input magnitude (lower right) shows that Wass-MD requires comparable control effort to other adaptive methods while achieving superior tracking performance.

**PVTOL: Underactuated System.** The underactuated PVTOL system presents a more challenging control problem due to the coupling between position and attitude dynamics. Figure 4 demonstrates that our Wass-MD controller significantly outperforms all baselines under time-varying wind conditions that ramp up to 8 m/s. The trajectory plot shows that both PID and MRR fail to maintain accurate tracking during the high-wind phase, while MLAC and Wass-MD successfully track the reference. Notably, the Lyapunov function $V(x(t), r(t))$ and position error $\|x(t) - r(t)\|_2$ plots reveal that Wass-MD achieves consistently lower values throughout the simulation, indicating superior stability margins and tracking accuracy.

**Parameter Evolution:** Figure 5 visualizes the evolution of adaptive parameters projected onto the first three principal components via PCA for the PFAR system. The Wass-MD controller exhibits a distinct trajectory in parameter space that converges more directly toward the final estimate compared to other methods. This behavior reflects the Wasserstein geometry's ability to guide parameter updates along transport-optimal paths rather than Euclidean shortest paths. Figure 6 compares the Lyapunov function and position error evolution for the PVTOL system. The Wass-MD controller maintains consistently lower Lyapunov values and achieves faster convergence of tracking error, demonstrating that the learned Wasserstein geometry improves adaptation in underactuated systems where the control authority is limited.

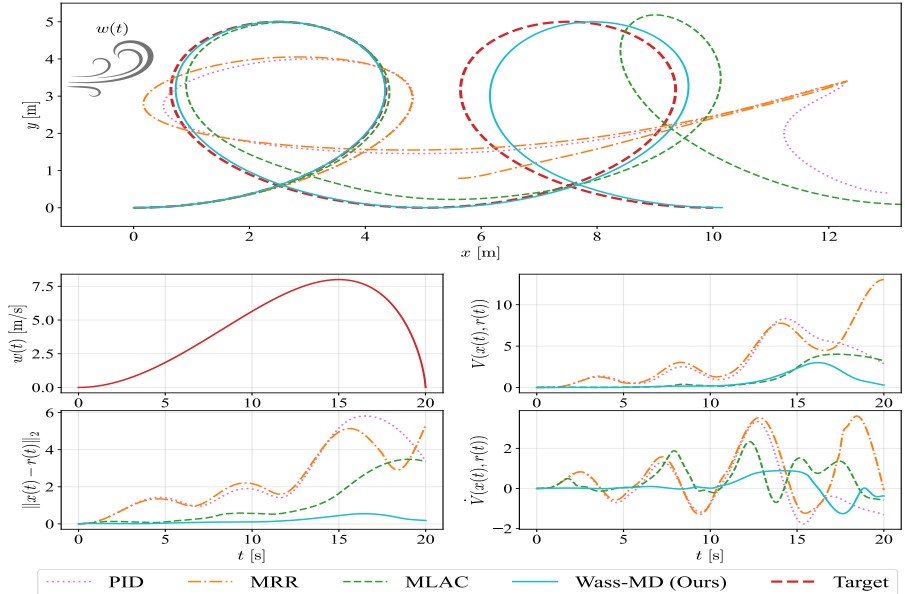

*Figure 4.* Tracking results for the PVTOL system on a double "loop-the-loop" with a time-varying wind velocity w(t).

**Computational Considerations.** The meta-learning phase is performed offline. The online controller evaluates particle updates per control step, with the dominant cost being the kernel matrix computation scaling as $O(K^2 d)$. In our setup, each control step requires approximately 1-2 ms on a single CPU core (Intel i7, 2.9GHz) with JAX compilation, well within the 10 ms budget for 100Hz control. This clean separation between the heavy offline meta-training and the lightweight online particle update is what makes the framework deployable on resource-constrained platforms, and the $O(K^2 d)$ kernel cost is trivially parallelizable on GPU for higher-dimensional systems.

Our experiments demonstrate that Wasserstein geometry-aware adaptation consistently outperforms Euclidean and static $p$-norm geometries on both fully-actuated and under-actuated systems, while exhibiting robustness under distributional shift where test wind speeds exceed training conditions. Additional results across varying wind speeds are provided in Appendix F.

### 6.3. Ablation Study: Sensitivity to Particle Count $K$

The particle count $K$ controls the discretization fidelity of the parameter measure $\nu_{\hat{a}}$ in our Wasserstein gradient flow and is therefore the key hyperparameter of the base-learner. We study its effect on closed-loop tracking with the meta-trained Wass-MD controller on both the fully-actuated PFAR and the underactuated PVTOL systems; the meta-training ensemble size $M = 5$ is held fixed.

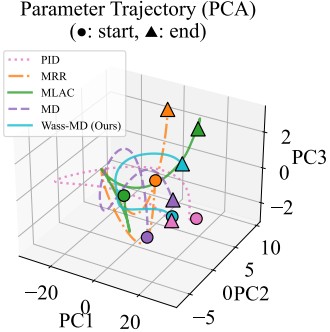

*Figure 5.* Adaptive parameter trajectories in principal component space (PFAR). PC1, PC2, and PC3 are the three principal component directions after performing PCA on the high-dimensional adaptive parameter matrix A, and represent the main change modes of parameter evolution.

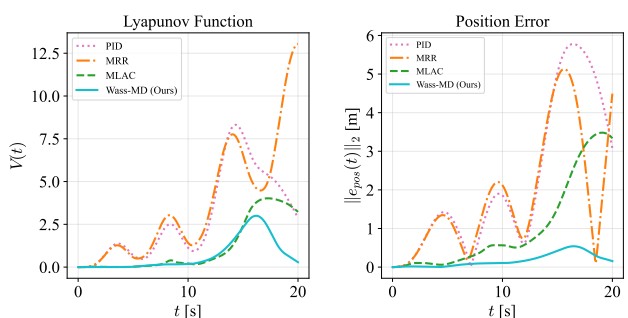

*Figure 6.* Lyapunov stability and tracking performance under time-varying wind (PVTOL)

Figures 7 and 8 together with Table 1 show that tracking accuracy improves monotonically with $K$ on both systems,

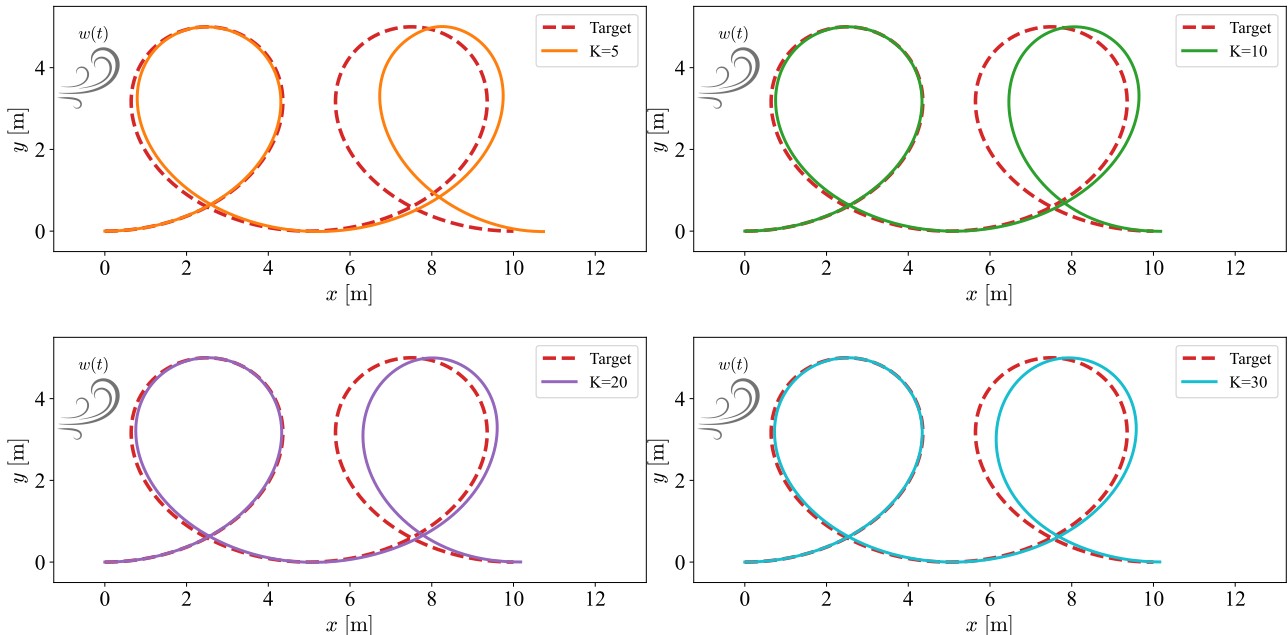

*Figure 7.* PVTOL tracking on a double loop-the-loop with time-varying wind $w(t)$ peaking at 8.0 m/s, for $K \in \{5, 10, 20, 30\}$. Larger $K$ tightens the tracking, with the most pronounced gains during the high-wind phase.

| $K$ | PFAR RMS [m] | PVTOL RMS [m] | Runtime/step |
|-----|--------------|---------------|--------------|
| 5   | 0.1242       | 0.2544        | $\sim$1.5 ms |
| 10  | 0.1125       | 0.2279        | $\sim$3 ms   |
| 20  | 0.0963       | 0.1935        | $\sim$6 ms   |
| 30  | 0.0546       | 0.1659        | $\sim$8 ms   |

*Table 1.* Tracking RMS error and online per-step runtime as a function of particle count $K$ (PFAR: $w = 8.0$ m/s; PVTOL: peak wind 8.0 m/s).

with PFAR RMS dropping by 56% from $K = 5$ to $K = 30$ and PVTOL RMS by 35%. The default $K = 5$ used elsewhere in this paper already attains competitive performance, while larger $K$ tightens tracking further at a modest runtime cost that stays compatible with 100 Hz control even at $K = 30$.

## 7. Conclusions

In this paper, we propose a Wasserstein geometry-aware adaptive control framework that leverages meta-learning to jointly optimize nonlinear feature representations, control gains, and transport geometry. Unlike existing approaches that rely on static Euclidean or $p$-norm geometries, our method constructs a data-driven Wasserstein Bregman divergence with meta-learned transport geometry, enabling the adaptation law to follow optimal transport paths for parameter updates. Closed-loop tracking simulations demonstrate that our controller achieves optimal performance on both fully-actuated and underactuated nonlinear planar rotorcraft,

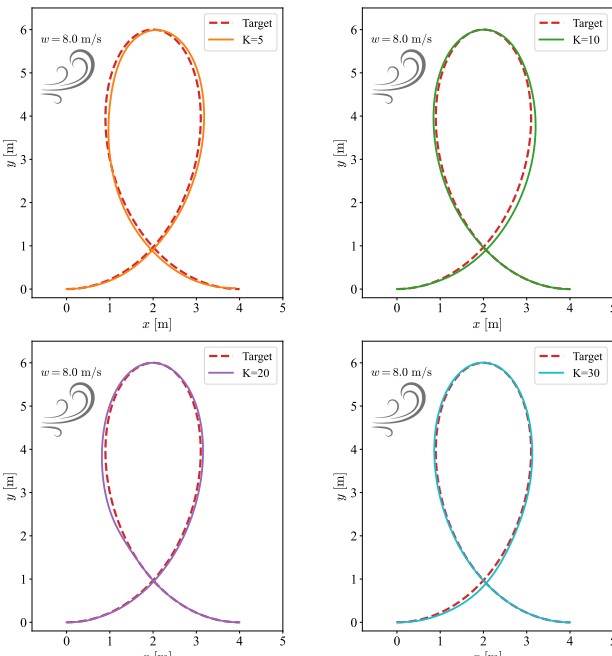

*Figure 8.* PFAR tracking under wind $w = 8.0$ m/s with varying particle counts $K \in \{5, 10, 20, 30\}$. Trajectories converge monotonically toward the reference as $K$ grows.

maintaining robustness under significant distributional shifts between training and testing conditions. Ablations on the particle count $K$ further confirm that $K = 5$ suffices in real time, with monotonic gains for larger $K$.

## Acknowledgements

This work was supported by the Shaanxi Provincial Scientists + Engineers Scientific Research Project 2024QCY-KXJ-173, the Key Research and Development Program of Shaanxi Province 2024PT-ZCK-77, and in part by the National Natural Science Foundation of China under Grant 62573340 and the National Science and Technology Major Projects under Grant 2025ZD0805901.

## Impact Statement

This paper presents work whose goal is to advance the field of Machine Learning. There are many potential societal consequences of our work, none which we feel must be specifically highlighted here.

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

# A. Background on Optimal Transport

In this section, we provide the necessary background on Optimal Transport (OT) and the geometry of the Wasserstein space that underpins our proposed adaptation framework. We focus on the dynamic formulation of OT and the associated Riemannian structure, which allows us to lift gradient-based adaptation laws from Euclidean space to the space of probability distributions.

## A.1. The Wasserstein Space

Let $\mathcal{P}_2(\mathbb{R}^d)$ denote the space of probability measures on $\mathbb{R}^d$ with finite second moments. The Wasserstein-2 distance between two measures $\mu, \nu \in \mathcal{P}_2(\mathbb{R}^d)$ is defined by the Kantorovich problem (Villani, 2008; Peyré & Cuturi, 2019):

$$W_2^2(\mu, \nu) = \inf_{\gamma \in \Pi(\mu, \nu)} \int_{\mathbb{R}^d \times \mathbb{R}^d} \|x - y\|^2 \, d\gamma(x, y), \tag{20}$$

where $\Pi(\mu, \nu)$ is the set of couplings (joint distributions) $\gamma$ on $\mathbb{R}^d \times \mathbb{R}^d$ such that the marginals are $\mu$ and $\nu$, respectively. When $\mu$ is absolutely continuous with respect to the Lebesgue measure, the optimal coupling is unique and induced by a transport map $T : \mathbb{R}^d \to \mathbb{R}^d$ such that $\nu = T_{\#}\mu$, known as the Monge map. This map satisfies $W_2^2(\mu, \nu) = \int \|x - T(x)\|^2 d\mu(x)$ (Villani, 2008).

## A.2. Riemannian Geometry and Tangent Space

The metric space $(\mathcal{P}_2(\mathbb{R}^d), W_2)$ possesses a formal Riemannian structure introduced by Otto (2001). The tangent space $T_\mu \mathcal{P}_2(\mathbb{R}^d)$ at a measure $\mu$ is identified with the closure of the set of gradients of smooth functions in $L^2(\mu)$ (Ambrosio et al., 2005):

$$T_\mu \mathcal{P}_2(\mathbb{R}^d) = \overline{\{\nabla \varphi : \varphi \in C_c^\infty(\mathbb{R}^d)\}}^{L^2(\mu)}. \tag{21}$$

This identification allows us to define the Riemannian metric tensor. For two tangent vectors $v_1, v_2 \in T_\mu \mathcal{P}_2(\mathbb{R}^d)$, the inner product is given by:

$$\langle v_1, v_2 \rangle_\mu = \int_{\mathbb{R}^d} \langle v_1(x), v_2(x) \rangle \, d\mu(x). \tag{22}$$

In this geometry, the exponential map $\exp_\mu : T_\mu \mathcal{P}_2(\mathbb{R}^d) \to \mathcal{P}_2(\mathbb{R}^d)$ is defined as $\exp_\mu(v) = (Id + v)_{\#}\mu$, where $Id$ is the identity map (Ambrosio et al., 2005).

## A.3. Wasserstein Gradient Flows

Consider a functional $\mathcal{F} : \mathcal{P}_2(\mathbb{R}^d) \to \mathbb{R}$. The Wasserstein gradient $\nabla_{W_2}\mathcal{F}(\mu)$ is the element of the tangent space $T_\mu \mathcal{P}_2(\mathbb{R}^d)$ that represents the steepest ascent direction of $\mathcal{F}$ according to the Wasserstein metric. It satisfies the first-order expansion (Jordan et al., 1998):

$$\mathcal{F}(\nu) = \mathcal{F}(\mu) + \int_{\mathbb{R}^d} \langle \nabla_{W_2}\mathcal{F}(\mu)(x), v(x) \rangle \, d\mu(x) + o(W_2(\mu, \nu)), \tag{23}$$

for $\nu = \exp_\mu(v)$. Formally, the Wasserstein gradient is related to the Euclidean gradient of the first variation $\frac{\delta \mathcal{F}}{\delta \mu}$ by $\nabla_{W_2}\mathcal{F}(\mu) = \nabla\left(\frac{\delta \mathcal{F}}{\delta \mu}\right)$. A Wasserstein gradient flow is a curve $\mu_t$ satisfying the continuity equation (Ambrosio et al., 2005):

$$\partial_t \mu_t + \text{div}(\mu_t v_t) = 0, \quad v_t = -\nabla_{W_2}\mathcal{F}(\mu_t). \tag{24}$$

## A.4. Particle Approximation

In our framework, we approximate the parameter distribution $\mu$ using a set of $N$ particles $\{\hat{a}^{(i)}\}_{i=1}^N$, corresponding to the empirical measure $\hat{\mu} = \frac{1}{K} \sum_{i=1}^K \delta_{\hat{a}^{(i)}}$. The Wasserstein gradient flow of a functional $\mathcal{F}$ on the space of measures translates to a system of coupled Ordinary Differential Equations (ODEs) on the particle positions. Specifically, if the functional $\mathcal{F}$ admits a closed form over discrete measures, the flow is discretized as:

$$\dot{\hat{a}}^{(i)} = -\nabla_{W_2}\mathcal{F}(\hat{\mu})(\hat{a}^{(i)}), \quad i = 1, \dots, K. \tag{25}$$

This particle-based perspective justifies the adaptation law derived in (16) of the main text, where the kernel interaction arises from the regularization of the Wasserstein gradient in the discrete setting (Bonet et al., 2024).

## B. Stability Analysis of Wasserstein Geometry-Aware Adaptive Control

In this appendix, we provide a rigorous stability analysis for the proposed Wasserstein geometry-aware adaptive controller. We establish that the closed-loop system achieves uniformly ultimately bounded tracking error.

First, we aim to make the system state $x(t)$ track a desired target trajectory $r(t)$, where $(r, u(r))$ is a known state–input pair that is nominally open-loop feasible, i.e., it satisfies the disturbance-free dynamics $\dot{x} = f(x) + g(x)u$.

Consider a scalar function $\bar{V}(x, r)$, where $\bar{V} : \mathbb{R}^n \times \mathbb{R}^n \to \mathbb{R}$. Taking its time derivative along the trajectories of the nominal system yields

$$\dot{\bar{V}}(x, r) = \nabla_x \bar{V}(x, r)^\top \big( f(x) + g(x)\bar{u} \big) + \nabla_r \bar{V}(x, r)^\top \big( f(r) + g(r)u(r) \big). \tag{26}$$

We seek to design a stabilizing feedback policy $\bar{u}(x, r)$ such that any trajectory $x(t)$ of the closed-loop nominal system $\dot{x} = f(x) + g(x)\bar{u}$ is guaranteed to converge to the reference trajectory $r(t)$, as certified by the accompanying function $\bar{V}(x, r)$.

Due to the complexity of nonlinear dynamics, deriving a closed-form expression for $\bar{u}$, a tracking controller for the nominal system, is generally challenging. In practice, controller design often relies on the specific structure of the dynamics as well as available computational resources. Representative approaches for constructing such tracking controllers include nonlinear control techniques such as control-contraction-metrics-based feedback (Manchester & Slotine, 2017), sliding mode control (Slotine & Li, 1991), and model predictive control (Qin & Badgwell, 2003). Despite their apparent diversity, these methods are unified by the existence of an associated Lyapunov-like scalar function $\bar{V}(x, r)$, which guarantees asymptotic stability of the tracking error.

Building on this discussion, we leverage the nominal pair $(\bar{u}, \bar{V})$ and subsequently augment it to achieve adaptive stabilization of the disturbed system described in (1).

Consider the uncertain nonlinear system (1) with the adaptive controller defined by:

$$u = \bar{u} - Y(x)\hat{a}, \tag{27}$$

where $\hat{a} = \frac{1}{K} \sum_{k=1}^{K} \hat{a}^{(k)}$ is the empirical mean of the particle estimates. The particle dynamics follow the Wasserstein gradient flow:

$$\dot{\hat{a}}^{(i)} = \sum_{j=1}^{K} \kappa_\sigma(\hat{a}^{(i)}, \hat{a}^{(j)}) \cdot \frac{1}{K} \cdot \mathcal{H}(\hat{a}^{(j)}), \tag{28}$$

where $\kappa_\sigma(a, b) = \exp(-\|a - b\|^2 / 2\sigma^2)$ is the Gaussian kernel with bandwidth $\sigma > 0$, and the mirror descent direction is:

$$\mathcal{H}(\hat{a}) = -P^{-1}(\nabla^2 \chi(P\hat{a}))^{-1} P^{-1} Y(x)^\top g(x)^\top \nabla_x \bar{V}(x, r). \tag{29}$$

We consider the Lyapunov-like function:

$$V(x, r, \nu_a, \nu_{\hat{a}}) = \bar{V}(x, r) + d_{\Psi_\mu}(P_\# \nu_a \| P_\# \nu_{\hat{a}}), \tag{30}$$

where $\nu_a = \delta_a$ is the Dirac measure at the true parameter $a$, and $\nu_{\hat{a}} \approx \hat{\nu}_K = \frac{1}{K} \sum_{k=1}^{K} \delta_{\hat{a}^{(k)}}$ is the empirical measure over particles.

For the empirical measure approximation (Bonet et al., 2024), the Wasserstein Bregman divergence admits the following representation:

$$d_{\Psi_\mu}(P_\# \delta_a \| P_\# \hat{\nu}_K) = \frac{1}{K} \sum_{k=1}^{K} d_\chi(Pa \| P\hat{a}^{(k)}), \tag{31}$$

where $d_\chi(\cdot \| \cdot)$ is the standard Bregman divergence induced by the potential $\chi$:

$$d_\chi(Pa \| P\hat{a}^{(k)}) = \chi(Pa) - \chi(P\hat{a}^{(k)}) - \langle \nabla \chi(P\hat{a}^{(k)}), Pa - P\hat{a}^{(k)} \rangle. \tag{32}$$

**Lemma B.1** (Decomposition of Wasserstein Bregman Divergence). *For the Dirac measure $\nu_a = \delta_a$ and empirical measure $\hat{\nu}_K = \frac{1}{K} \sum_{k=1}^{K} \delta_{\hat{a}^{(k)}}$, the following identity holds:*

$$d_{\Psi_\mu}(P_\# \delta_a \| P_\# \hat{\nu}_K) = \frac{1}{K} \sum_{k=1}^{K} d_\chi(Pa \| P\hat{a}^{(k)}). \tag{33}$$

*Proof.* By definition of the potential functional (8):

$$\Psi_\mu(P_\# \delta_a) = \int_{\mathbb{R}^d} \chi(b) \, d(P_\# \delta_a)(b) = \chi(Pa), \tag{34}$$

$$\Psi_\mu(P_\# \hat{\nu}_K) = \int_{\mathbb{R}^d} \chi(b) \, d(P_\# \hat{\nu}_K)(b) = \frac{1}{K} \sum_{k=1}^{K} \chi(P\hat{a}^{(k)}). \tag{35}$$

The first variation of $\Psi_\mu$ at $P_\# \hat{\nu}_K$ evaluated at a test measure $\mu$ is $\langle \nabla_\mu \Psi_\mu(P_\# \hat{\nu}_K), \mu \rangle = \int \chi(b) \, d\mu(b)$. Substituting into the Wasserstein Bregman divergence definition (9) and using the convexity structure yields the result. $\square$

For each particle $k$, the time derivative of the Bregman divergence satisfies:

$$\begin{aligned}
\frac{d}{dt} d_\chi(Pa \| P\hat{a}^{(k)}) &= \frac{d}{dt} \left[ \chi(Pa) - \chi(P\hat{a}^{(k)}) - \nabla\chi(P\hat{a}^{(k)})^\top (Pa - P\hat{a}^{(k)}) \right] \\
&= -\nabla\chi(P\hat{a}^{(k)})^\top P\dot{\hat{a}}^{(k)} - \nabla^2\chi(P\hat{a}^{(k)}) P\dot{\hat{a}}^{(k)} \cdot (Pa - P\hat{a}^{(k)}) \\
&\quad + \nabla\chi(P\hat{a}^{(k)})^\top P\dot{\hat{a}}^{(k)} \\
&= -(Pa - P\hat{a}^{(k)})^\top \nabla^2\chi(P\hat{a}^{(k)}) P\dot{\hat{a}}^{(k)} \\
&= -\tilde{a}^{(k)\top} P^\top \nabla^2\chi(P\hat{a}^{(k)}) P\dot{\hat{a}}^{(k)}.
\end{aligned} \tag{36}$$

where $\tilde{a}^{(k)} = a - \hat{a}^{(k)}$ is the parameter estimation error for the $k$-th particle.

**Lemma B.2.** *Under the Wasserstein Geometry-Aware Adaptive Control, consider the Lyapunov-like function defined in (10):*

$$V(x, r, \nu_a, \nu_{\hat{a}}) = \bar{V}(x, r) + d_{\Psi_\mu}(P_\# \nu_a \| P_\# \nu_{\hat{a}}), \tag{37}$$

*where $P = P^\top \succ 0$ and $\Psi_\mu$ is the potential functional defined in (8). Then, the closed-loop system is stable, and the tracking error converges to a compact set whose size is determined by the kernel bandwidth $\sigma$.*

*Proof.* First, we analyze the derivative of the nominal Lyapunov candidate $\bar{V}(x, r)$ under the adaptive control law $u = \bar{u} - Y(x)\hat{a}$, where $\hat{a} = \frac{1}{K} \sum_{k=1}^{K} \hat{a}^{(k)}$ is the particle mean. The closed-loop dynamics are $\dot{x} = f(x) + g(x)\bar{u} + g(x)Y(x)(a - \hat{a})$. Thus:

$$\dot{\bar{V}} = \nabla_x \bar{V}(x, r)^\top \big( f(x) + g(x)\bar{u} \big) + \nabla_r \bar{V}(x, r)^\top \big( f(r) + g(r)u(r) \big) + \nabla_x \bar{V}(x, r)^\top g(x)Y(x)(a - \hat{a}). \tag{38}$$

To simplify notation, we define the regressor vector $\Phi(x)$ as:

$$\Phi(x) := Y(x)^\top g(x)^\top \nabla_x \bar{V}(x, r). \tag{39}$$

The disturbance term can then be written as $\Phi(x)^\top (a - \hat{a}) = \frac{1}{K} \sum_{k=1}^{K} (a - \hat{a}^{(k)})^\top \Phi(x)$.

Next, to facilitate the geometric analysis, let $M(z) = P^\top \nabla^2 \chi(Pz) P$ denote the Riemannian metric tensor induced by the potential $\chi$. Combining the nominal derivative with the time derivative of the Wasserstein divergence, we obtain the total derivative expression:

$$\begin{aligned}
\dot{V} = {}& \nabla_x \bar{V}(x, r)^\top \big( f(x) + g(x)\bar{u} \big) + \nabla_r \bar{V}(x, r)^\top \big( f(r) + g(r)u(r) \big) \\
& + \frac{1}{K} \sum_{k=1}^{K} (a - \hat{a}^{(k)})^\top \Phi(x) - \frac{1}{K} \sum_{k=1}^{K} (a - \hat{a}^{(k)})^\top M(\hat{a}^{(k)}) \dot{\hat{a}}^{(k)}.
\end{aligned} \tag{40}$$

Substituting the Wasserstein gradient flow update law $\dot{\hat{a}}^{(k)} = \frac{1}{K} \sum_{j=1}^{K} \kappa_{kj} M(\hat{a}^{(j)})^{-1} \Phi(x)$, where $\kappa_{kj} := \kappa_\sigma(\hat{a}^{(k)}, \hat{a}^{(j)})$, the last term in (40) becomes a double sum. We introduce the geometric mismatch operator $\Delta_{kj}$ via the identity $M(\hat{a}^{(k)}) M(\hat{a}^{(j)})^{-1} = I + \Delta_{kj}$, where

$$\Delta_{kj} = \left[ M(\hat{a}^{(k)}) - M(\hat{a}^{(j)}) \right] M(\hat{a}^{(j)})^{-1}. \tag{41}$$

Substituting this back into (40) allows us to separate the ideal cancellation term from the geometric error:

$$
\begin{aligned}
\dot{V} = &\nabla_x \bar{V}(x,r)^\top \big( f(x) + g(x)\bar{u} \big) + \nabla_r \bar{V}(x,r)^\top \big( f(r) + g(r)u(r) \big) \\
&+ \frac{1}{K} \sum_{k=1}^{K} (a - \hat{a}^{(k)})^\top \Phi(x) \underbrace{\left( 1 - \frac{1}{K} \sum_{j=1}^{K} \kappa_{kj} \right)}_{\text{Density Error } \epsilon_{\text{dens}}^{(k)}} \\
&- \frac{1}{K^2} \sum_{k=1}^{K} \sum_{j=1}^{K} \kappa_{kj} (a - \hat{a}^{(k)})^\top \Delta_{kj} \Phi(x).
\end{aligned}
\tag{42}
$$

**Assumption B.3.** (Bonet et al., 2024) The Hessian $\nabla^2 \chi$ is Lipschitz continuous with constant $L_\chi$, and the metric inverse is bounded such that $\|M^{-1}\| \leq \beta$. This implies $\|\Delta_{kj}\| \leq L_\chi \beta \|\hat{a}^{(k)} - \hat{a}^{(j)}\|$.

**Assumption B.4.** (Bonet et al., 2024) For the Gaussian kernel, the interaction strength decays exponentially. The weighted local diameter is bounded by the bandwidth: $\sum_j \kappa_{kj} \|\hat{a}^{(k)} - \hat{a}^{(j)}\| \leq C_\sigma \cdot \sigma$.

Applying these bounds and Cauchy-Schwarz inequality to the error term:

$$\left| \frac{1}{K^2} \sum_{k,j} \kappa_{kj} (a - \hat{a}^{(k)})^\top \Delta_{kj} \Phi(x) \right| \leq \frac{\|\Phi(x)\|}{K} \sum_{k=1}^{K} \|a - \hat{a}^{(k)}\| \cdot (L_\chi \beta C_\sigma \sigma). \tag{43}$$

Thus,

$$\dot{V} \leq \nabla_x \bar{V}(x,r)^\top \big( f(x) + g(x)\bar{u} \big) + \nabla_r \bar{V}(x,r)^\top \big( f(r) + g(r)u(r) \big) + \|\Phi(x)\| \cdot (\mathcal{O}(\sigma) + \mathcal{O}(\epsilon_{\text{dens}})). \tag{44}$$

Consequently, this guarantees that the system is uniformly ultimately bounded, with the state $x(t)$ converging to a compact set around $r(t)$ whose size is controlled by the meta-learned geometry $\sigma$. $\qquad\square$

*Remark* B.5. The ultimate bound depends on the kernel bandwidth $\sigma$ through two mechanisms: (i) the geometric mismatch error scales as $\mathcal{O}(\sigma)$, and (ii) the density estimation error $\epsilon_{\text{dens}}^{(k)}$ depends on the particle spread relative to $\sigma$. In practice, the meta-learned $\sigma$ balances these effects to minimize the tracking error.

*Remark* B.6. When $K = 1$ (single particle) and $\sigma \to 0$, the density error $\epsilon_{\text{dens}} \to 0$. The controller reduces to the standard mirror descent adaptive controller of Tang et al. (2025), which achieves asymptotic tracking.

**Where the occupancy measure $\mu$ enters the framework.** We clarify the precise role of the occupancy measure $\mu$, since this distinction is important for interpreting the contribution correctly. For the Dirac measure $\nu_a = \delta_a$ and the empirical particle measure $\hat{\nu}_K$, the Wasserstein Bregman divergence in (10) admits a closed-form decomposition into an average of standard Bregman divergences (Lemma B.1), so $\mu$ does not appear *algebraically* inside the Lyapunov term once measures are specialized. Instead, $\mu$ enters the framework through the *meta-learning loop*: the geometry parameters $(p, \sigma)$ that govern the particle dynamics in (15) are optimized against the meta-objective, whose expectation is taken over the closed-loop trajectories that constitute $\mu$ via (11). Two different occupancy measures therefore yield two different meta-learned $(p^\star, \sigma^\star)$, and hence two different particle interaction kernels and transport geometries at deployment. In this sense, the Wasserstein lifting is realized *operationally*—through $\mu$-conditioned meta-training of the transport geometry—rather than as an explicit $\mu$-dependent term inside the stability certificate. The stability guarantee itself is uniformly ultimately bounded (UUB), with the ultimate bound shrinking as $\sigma \to 0$ and $K$ grows. As a consistency check, the degenerate limit $K = 1$, $\sigma \to 0$ recovers exactly the mirror-descent adaptive law of Tang et al. (2025) together with its asymptotic tracking guarantee.

## C. Additional Details on Model Ensembles

In practical scenarios where an exact oracle of the dynamics $f$ is unavailable, we operate in a data-driven setting using historical data collected by a behavior policy. For a shared set of sampled disturbance parameters $\{w_j\}_{j=1}^{M}$, we observe the corresponding state-action histories. The trajectory dataset for the $j$-th task is defined as:

$$\mathcal{T}_j = \left\{ \left( t_k^{(j)}, x_k^{(j)}, u_k^{(j)}, t_{k+1}^{(j)}, x_{k+1}^{(j)} \right) \right\}_{k=0}^{N_j - 1}, \tag{45}$$

where $x_k^{(j)}$ and $u_k^{(j)}$ denote the state and control input at time $t_k^{(j)}$, respectively.

To compensate for the unknown dynamics, we train a distinct surrogate model $\hat{f}_j(\cdot; \psi_j)$ for each trajectory $\mathcal{T}_j$. The parameters $\psi_j$ are optimized via minimizing the one-step prediction error using gradient descent:

$$\min_{\psi_j} \quad \frac{1}{N_j} \sum_{k=0}^{N_j - 1} \left\| x_{k+1}^{(j)} - \hat{x}_{k+1}^{(j)} \right\|_2^2 + \mu_{\text{ensem}} \|\psi_j\|_2^2$$

$$\text{s.t.} \quad \hat{x}_{k+1}^{(j)} = x_k^{(j)} + \int_{t_k^{(j)}}^{t_{k+1}^{(j)}} \hat{f}_j(x(\tau), u_j^{(k)}; \psi_j) \, d\tau. \tag{46}$$

The regularization term $\mu_{\text{ensem}} > 0$ prevents overfitting to limited trajectory data.

The resulting ensemble $\{\hat{f}_j\}_{j=1}^{M}$ serves as a proxy environment. This approach mitigates the inherent bias of individual estimators and enhances robustness against distributional shifts, a strategy validated in deep predictive modeling and reinforcement learning (Lakshminarayanan et al., 2017; Rajeswaran et al., 2017; Kurutach et al., 2018). Consequently, the meta-learning objective is reformulated to optimize the policy parameters $\theta$ over this learned ensemble:

$$\min_{\theta} \quad \frac{1}{NMT} \left( \sum_{i=1}^{N} \sum_{j=1}^{M} \int_0^T c_{ij}(t) \, dt + \mu_{\text{meta}} \|\theta\|_2^2 \right)$$

$$\begin{aligned} \text{s.t.} \quad & c_{ij} = \|x_{ij} - r_i\|^2 + \alpha \|u_{ij}\|_2^2, \\ & \dot{x}_{ij} = \hat{f}_j(x_{ij}, u_{ij}; \psi_j), \quad x_{ij}(0) = r_i(0), \\ & \dot{a}_{ij} = \rho(x_{ij}, r_i, \hat{a}_{ij}; \theta), \quad \hat{a}_{ij}(0) = 0, \\ & u_{ij} = u(x_{ij}, r_{ij}, \hat{a}_{ij}; \theta). \end{aligned} \tag{47}$$

Note that unlike standard model-based RL which often aggregates predictions, here each task-specific loss is computed using its corresponding surrogate $\hat{f}_j$, preserving the structure of the disturbance distribution.

## D. Stability Analysis for Lagrangian Systems

In this section, we apply the proposed Wasserstein geometric adaptive controller to the fundamental class of Lagrangian dynamical systems, which models complex mechanical systems such as robotic arms and quadrotors.

We define the system state as $x := (q, \dot{q})$, where $q(t) \in \mathbb{R}^d$ represents the generalized coordinates. The nonlinear dynamics are described by:

$$H(q)\ddot{q} + C(q, \dot{q})\dot{q} + g(q) = f_{\text{ext}}(q, \dot{q}) + \tau(u), \tag{48}$$

where $H(q)$ is the inertia matrix, $C(q, \dot{q})$ is the Coriolis/centrifugal matrix, and $g(q)$ is the gravity vector.

### D.1. Fully-Actuated Lagrangian Systems

The system is fully-actuated ($\tau$ is invertible). We assume $f_{\text{ext}}(q, \dot{q}) = Y(q, \dot{q})a$. The controller is defined as:

$$e := q - q_d, \quad s := \dot{e} + \Lambda e, \quad v := \dot{q}_d - \Lambda e,$$

$$u = \tau^{-1}\big(H(q)\dot{v} + C(q, \dot{q})v + g(q) - Y(q, \dot{q})\hat{a} - Ks\big),$$

$$\dot{\hat{a}}^{(k)} = \frac{1}{K} \sum_{j=1}^{K} \kappa_\sigma(\hat{a}^{(k)}, \hat{a}^{(j)}) \mathcal{H}(\hat{a}^{(j)}), \tag{49}$$

where $\mathcal{H}(\hat{a}^{(j)}) = -P^{-1}(\nabla^2\chi(P\hat{a}^{(j)}))^{-1}P^{-1}Y(q,\dot{q})^\top s$.

*Proof.* Let $\mathcal{N}(q, \dot{q}, v, \dot{v}) = H(q)\dot{v} + C(q, \dot{q})v + g(q)$ be the nominal dynamics required to follow the reference. The system dynamics in terms of the composite error $s = \dot{q} - v$ can be written as:

$$H(q)\dot{s} = -C(q, \dot{q})s - \mathcal{N} + f_{\text{ext}} + \tau(u). \tag{50}$$

Substituting the control law $u$, we obtain the closed-loop error dynamics:

$$H(q)\dot{s} = -(C(q, \dot{q}) + K)s + Y(q, \dot{q})(a - \hat{a}). \tag{51}$$

Consider the Lyapunov function $V = \frac{1}{2}s^\top Hs + d_{\Psi_\mu}(P_\#\nu_a\|P_\#\nu_{\hat{a}})$. Taking the time derivative:

$$
\begin{aligned}
\dot{V} &= s^\top H\dot{s} + \frac{1}{2}s^\top \dot{H}s + \frac{d}{dt}d_{\Psi_\mu}(P_\#\nu_a\|P_\#\nu_{\hat{a}}) \\
&= s^\top\left(-(C + K)s + Y(a - \hat{a})\right) + \frac{1}{2}s^\top \dot{H}s + \frac{d}{dt}d_{\Psi_\mu}(P_\#\nu_a\|P_\#\nu_{\hat{a}}).
\end{aligned}
\tag{52}
$$

Using the skew-symmetry property $s^\top(\dot{H} - 2C)s = 0$, this simplifies to:

$$\dot{V} = -s^\top Ks + s^\top Y(a - \hat{a}) + \frac{d}{dt}d_{\Psi_\mu}(P_\#\nu_a\|P_\#\nu_{\hat{a}}). \tag{53}$$

Substituting the Wasserstein gradient update law:

$$s^\top Y(a - \hat{a}) + \frac{d}{dt}d_{\Psi_\mu}(P_\#\nu_a\|P_\#\nu_{\hat{a}}) \le \|s\|\|Y\|\left(\mathcal{O}(\epsilon_{\text{dens}}) + \mathcal{O}(\sigma)\right). \tag{54}$$

Thus, the total derivative is bounded by:

$$\dot{V} \le -\lambda_{\min}(K)\|s\|^2 + \|s\| \cdot \|Y\| \cdot \left(\mathcal{O}(\epsilon_{\text{dens}}) + \mathcal{O}(\sigma)\right). \tag{55}$$

This proves the system is uniformly ultimately bounded. $\square$

### D.2. Underactuated Lagrangian Systems

For underactuated systems ($m < d$), the input maps through $B(q) \in \mathbb{R}^{d\times m}$. The dynamics are:

$$H(q)\ddot{q} + C(q, \dot{q})\dot{q} + g(q) = f_{\text{ext}}(q, \dot{q}) + B(q)u. \tag{56}$$

We assume the disturbance consists of a matched component $f_{\text{matched}} = B(q)Y(q, \dot{q})a$ and a residual. The controller uses the pseudoinverse $B^\dagger = (B^\top B)^{-1}B^\top$:

$$u = B^\dagger(q)\left(\mathcal{N}(q, \dot{q}, v, \dot{v}) - B(q)Y(q, \dot{q})\hat{a} - Ks\right). \tag{57}$$

**Assumption D.1.** (Hauser et al., 1992) The internal dynamics of the underactuated system are assumed to be Input-to-State Stable (ISS) with respect to the tracking error $s$. This implies that the unactuated states remain uniformly bounded provided that the tracking error $s$ is bounded.

*Proof.* First, we explicitly derive the closed-loop dynamics. Let $P_B = BB^\dagger$ be the orthogonal projection onto the actuated subspace, and let $P_B^\perp = I - P_B$ be the projection onto the unactuated (null) subspace. The open-loop error dynamics are:

$$H\dot{s} = -Cs - \mathcal{N} + f_{\text{ext}} + Bu. \tag{58}$$

Substituting the control input $u$, and noting that $BB^\dagger = P_B$:

$$
\begin{aligned}
Bu &= P_B\left(\mathcal{N} - BY\hat{a} - Ks\right) \\
&= P_B\mathcal{N} - BY\hat{a} - P_BKs.
\end{aligned}
\tag{59}
$$

Substituting this back into the error dynamics:

$$H\dot{s} = -Cs - \mathcal{N} + f_{\text{ext}} + (P_B\mathcal{N} - BY\hat{a} - P_BKs). \tag{60}$$

Note that $-\mathcal{N} + P_B\mathcal{N} = -(I - P_B)\mathcal{N} = -P_B^{\perp}\mathcal{N}$. Similarly, we decompose the external disturbance into matched and unmatched components: $f_{\text{ext}} = BYa + f_{\text{ext}}^{\perp}$.

$$H\dot{s} = -(C + P_BK)s + (BYa - BY\hat{a}) + \underbrace{f_{\text{ext}}^{\perp} - P_B^{\perp}\mathcal{N}}_{\Delta_{\text{total}}}. \tag{61}$$

Here, $\Delta_{\text{total}} := (I - P_B)(f_{\text{ext}} - \mathcal{N})$ represents the total unmatched dynamics. This term captures forces that physically cannot be cancelled by the actuators and the component of the nominal dynamics that excites the unactuated degrees of freedom.

Using the same Lyapunov function as before, the time derivative is:

$$\dot{V} = -s^{\top}P_BKs + s^{\top}BY(a - \hat{a}) + s^{\top}\Delta_{\text{total}} + \frac{d}{dt}d_{\Psi_\mu}(P_{\#}\nu_a\|P_{\#}\nu_{\hat{a}}). \tag{62}$$

Combining these bounds yields the final stability inequality:

$$\dot{V} \leq -\lambda_{\min}^{+}(P_BK)\|s\|^2 + \|s\|\left(\underbrace{\|BY\|(\mathcal{O}(\epsilon_{\text{dens}}) + \mathcal{O}(\sigma))}_{\text{Geometric Error}} + \underbrace{\|\Delta_{\text{total}}\|}_{\text{Unmatched Dynamics}}\right). \tag{63}$$

For sufficiently large $\|s\|$, the quadratic damping term dominates the linear disturbance terms, ensuring $\dot{V} < 0$. Thus, the error $s(t)$ converges to a compact set whose size is determined by the kernel bandwidth $\sigma$ and the magnitude of the underactuated dynamics $\Delta_{\text{total}}$. □

*Remark* D.2. The stability analysis in (62) accounts for the coupling between the tracking error $s$ and the unmatched dynamics $\Delta_{\text{total}}$. Crucially, the ISS assumption implies that $\Delta_{\text{total}}$ is bounded by a class $\mathcal{K}$ function of $\|s\|$. Since the Lyapunov derivative provides quadratic damping $(-\|s\|^2)$ that dominates this coupling effect for large errors, the coupled system is guaranteed to be uniformly ultimately bounded. This argument ensures that the controller effectively suppresses the disturbance without exciting unstable internal dynamics.

## E. Benchmark with Baseline Method

We compare our meta-trained Wasserstein Mirror-Descent-based adaptive controller in trajectory tracking tasks for (17) against three baseline controllers:

PID Control: Our simplest baseline is a ProportionalIntegral-Derivative (PID) controller with feed-forward, i.e.,

$$u = R(\phi)^{\top}\left(g + \ddot{q}_d - K_Pe - K_I\int_0^t e(\epsilon)\,d\epsilon - K_D\dot{e}\right), \tag{64}$$

with gains $K_P, K_I, K_D \succ 0$. For $f_{\text{ext}}(q, \dot{q}, w) \equiv 0$, this controller feedback-linearizes the dynamics (17) such that the error $e := q - q_d$ is governed by an exponentially stable ODE. The integral term must compensate for $f_{\text{ext}}(q, \dot{q}, w) \neq 0$.

Meta-Learning Adaptive Control (MLAC): The base-learner is an adaptive controller with GD-based adaptation laws, and the meta-learner includes the control gains $(\Lambda, K, \Gamma)$ and feature-function neural-network parameters $\theta_y$,

$$u = \tau^{-1}\left(H(q)\dot{v} + C(q, \dot{q})v + g(q) - y(q, \dot{q}; \theta_y)\hat{a} - Ks\right),$$
$$\dot{\hat{a}} = \Gamma y(q, \dot{q}; \theta_y)^{\top}s \tag{65}$$

Meta-trained MD-based Adaptive Control(MD): The base-learner is an adaptive controller with MD-based adaptation laws, and the meta-learner includes the control gains $(\Lambda, K, P, p)$ and feature-function neural-network parameters $\theta_y$,

$$u = \tau^{-1}\left(H(q)\dot{v} + C(q, \dot{q})v + g(q) - y(q, \dot{q}; \theta_y)\hat{a} - Ks\right),$$
$$\dot{\hat{a}} = -P^{-1}(\nabla^2\chi(P\hat{a}))^{-1}P^{-1}y(q, \dot{q}; \theta_y)^{\top}s \tag{66}$$

In comparison, our proposed method in this paper uses an adaptive controller with Wasserstein MD-based adaptation laws as the base-learner and leverages the meta-learner to search for an optimal $p$ in Wasserstein space which determines the potential function from the family $\chi(a) = \|a\|_p^p$.

Meta-Ridge Regression (MRR): Ridge regression is used as a base-learner to meta-learn parametric features $y(x; \theta_y)$ (Bertinetto et al., 2019; O'Connell et al., 2022). Given a trajectory of data $\mathcal{T}_j$, let $\mathcal{K}_j^{\text{ridge}} \subset \{k\}_{k=0}^{|\mathcal{T}_j|-1}$ denote the indices of transition tuples in some subset of $\mathcal{T}_j$. Define $\Delta t_k^{(j)} := t_{k+1}^{(j)} - t_k^{(j)}$, and the Euler approximation

$$\hat{x}_{k+1}^{(j)}(\hat{a}) := x_k^{(j)} + \Delta t_k^{(j)}\Big(f\big(x_k^{(j)}\big) + g\big(x_k^{(j)}\big)u_k^{(j)}\Big) \\ + \Delta t_k^{(j)}\Big(g\big(x_k^{(j)}\big)\hat{a}y\big(x_k^{(j)}; \theta_y\big)\Big) \tag{67}$$

as a function of $\hat{a}$. MRR posits that $\hat{a}$ should fit some subset of the trajectory in a regression sense; we can express this with the adaptation mechanism

$$\hat{a}_j = \arg \min_{\hat{a} \in \mathbb{R}^{m \times p}} \sum_{k \in \mathcal{K}_j^{\text{ridge}}} \left\| \hat{x}_{k+1}^{(j)}(\hat{a}) - x_{k+1}^{(j)} \right\|_2^2 + \mu_{\text{ridge}} \|a\|_F^2. \tag{68}$$

The task loss associated with trajectory $\mathcal{T}_j$ is then the regression loss

$$\mathcal{L}_j(\hat{a}_j, \mathcal{T}_j) = \frac{1}{|\mathcal{T}_j|} \sum_{k=0}^{|\mathcal{T}_j|-1} \left\| \hat{x}_{k+1}^{(j)}(\hat{a}) - x_{k+1}^{(j)} \right\|_2^2. \tag{69}$$

The meta-problem for MRR takes the form of (3) with features $y(x; \theta_y)$, the task loss (69), and the adaptation mechanism (68). The meta-parameters $\theta_{\text{MRR}} := \theta_y$ are trained via gradient descent on this meta-problem, and then deployed online via the features $y(q, \dot{q}; \theta_y)$ in the adaptive controller (65).

## F. Additional results

Additional results across wind speeds ranging from $2 - 10 m/s$ are presented below. As the disturbance magnitude increases, baseline controllers exhibit pronounced performance degradation, characterized by increased oscillations. In contrast, the proposed Wass-MD controller consistently maintains stable and accurate trajectory tracking across all wind conditions. Even at $10 m/s$, which lies well beyond the training range, Wass-MD achieves lower tracking errors and smoother control responses than all competing methods. These results confirm the robustness and superior generalization capability of the proposed approach under severe and out-of-distribution disturbances.

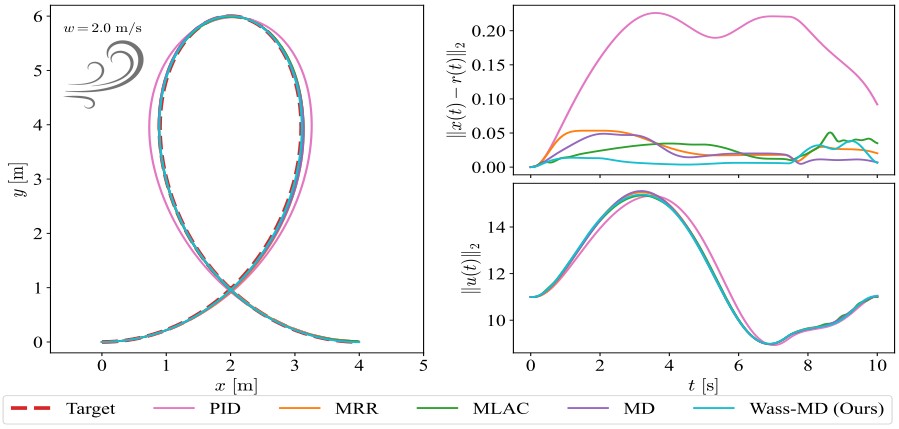

*Figure 9.* Tracking results for the PFAR on a test trajectory with w = 2.0 m/s

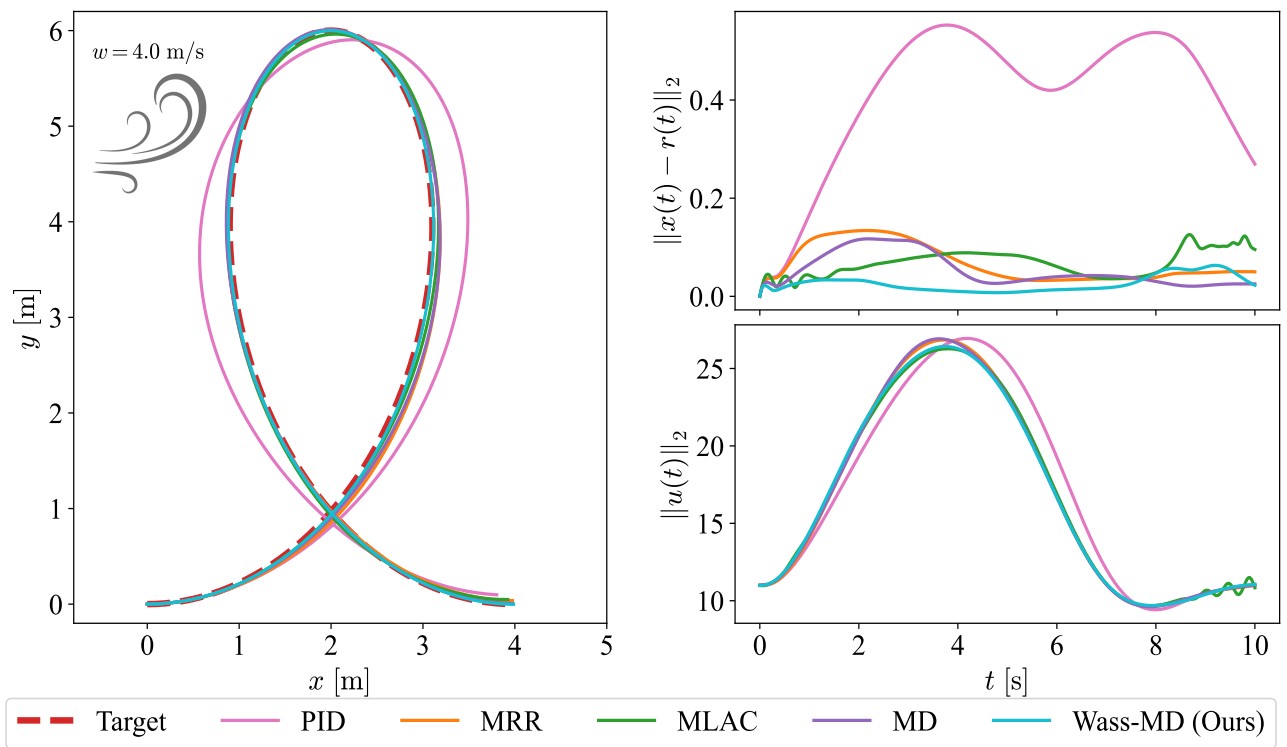

*Figure 10.* Tracking results for the PFAR on a test trajectory with w = 4.0 m/s

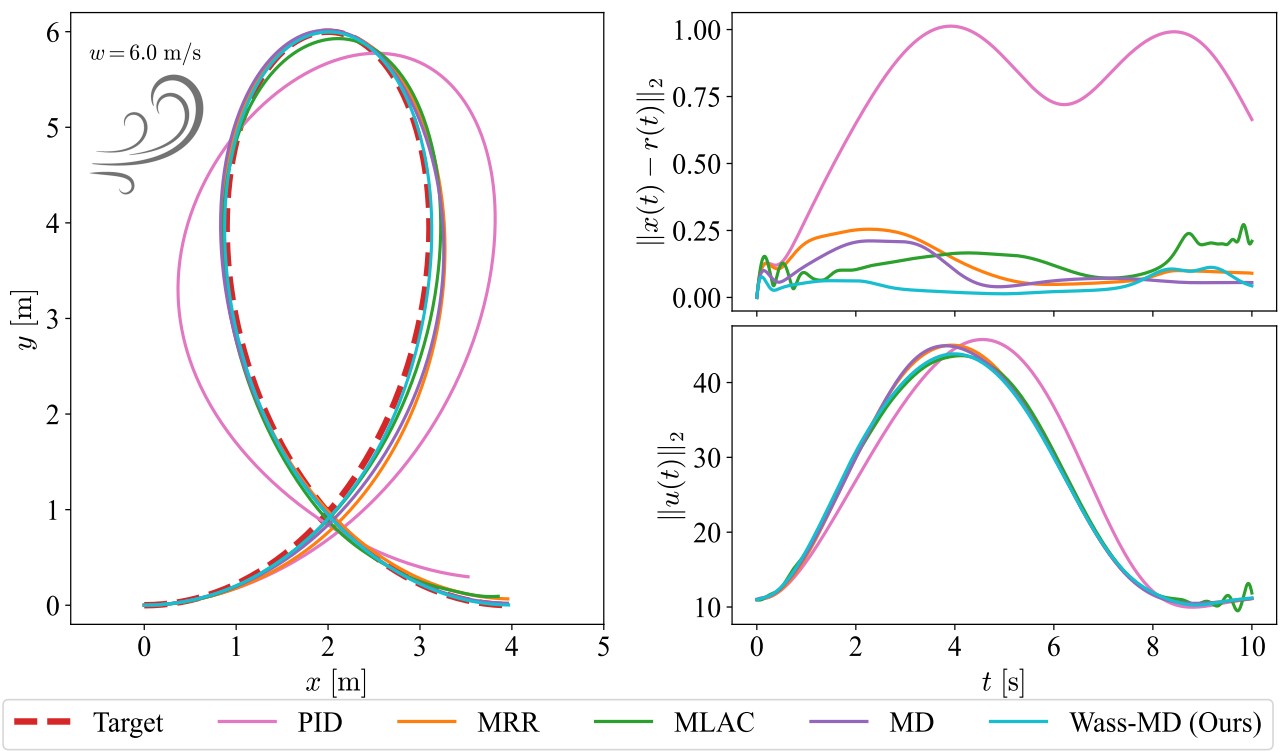

*Figure 11.* Tracking results for the PFAR on a test trajectory with w = 6.0 m/s

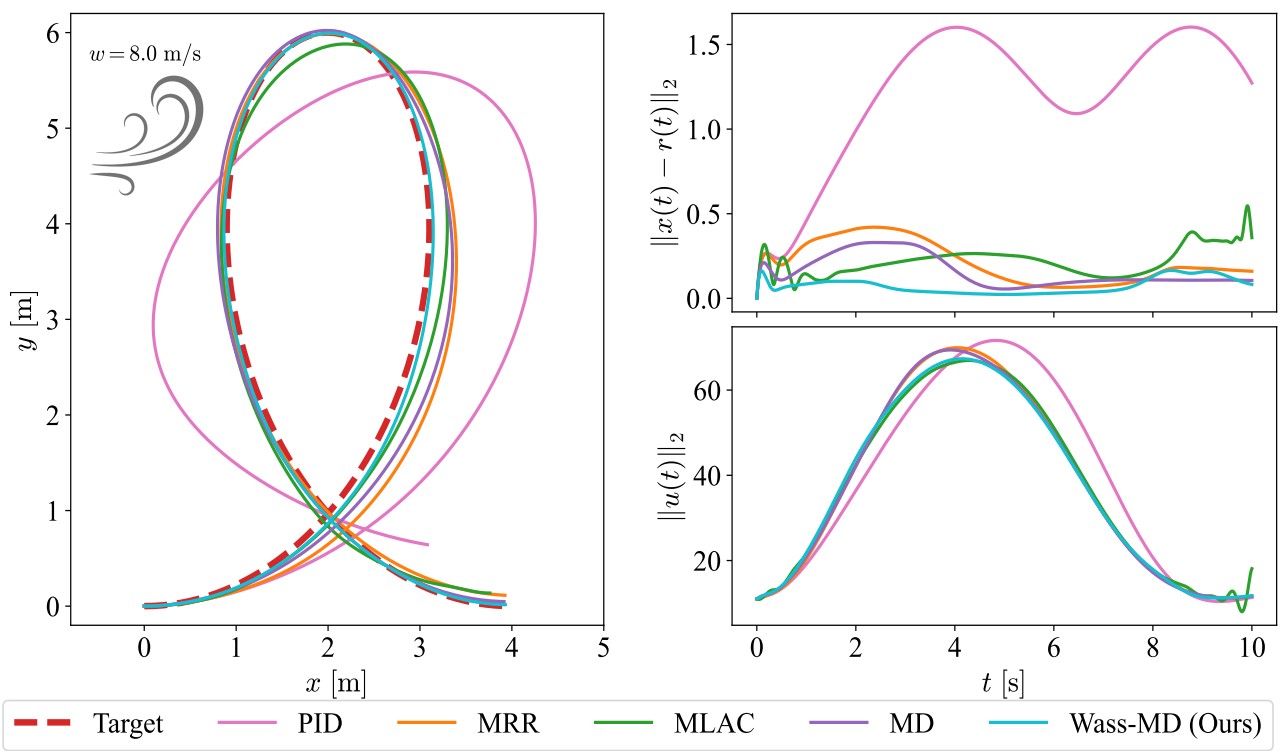

*Figure 12.* Tracking results for the PFAR on a test trajectory with w = 8.0 m/s

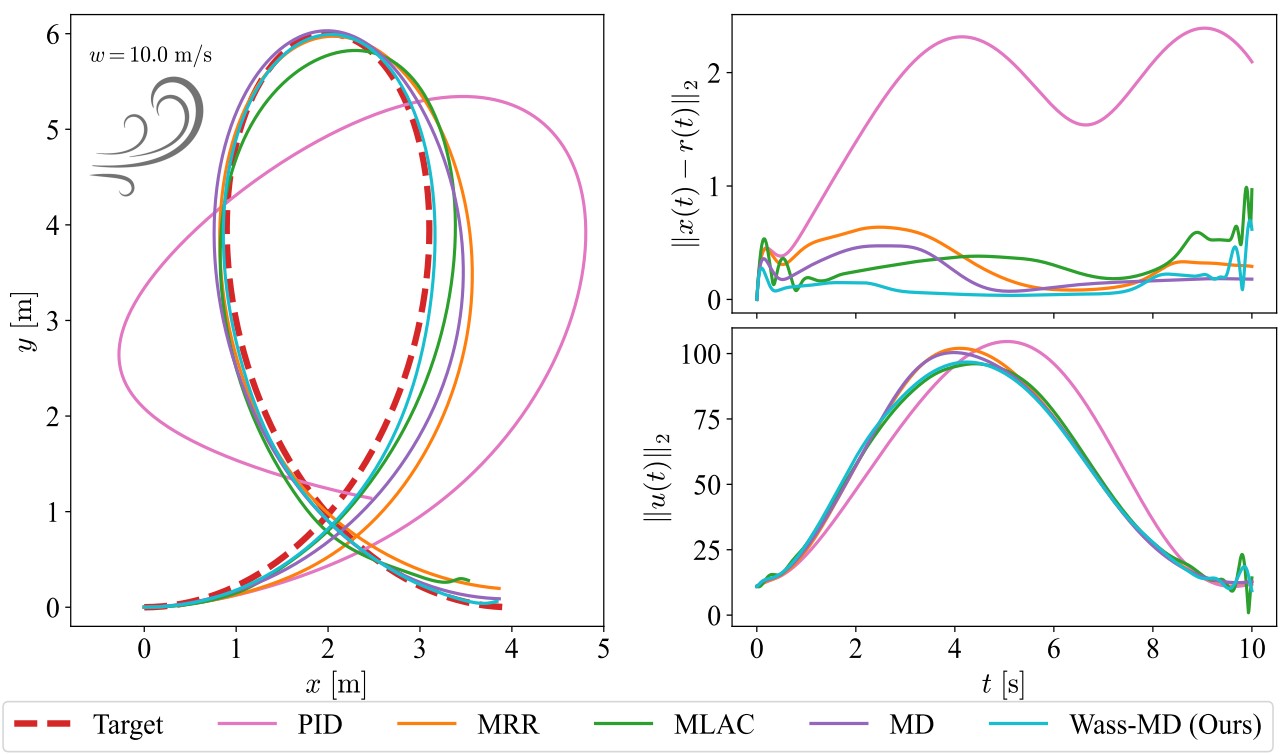

*Figure 13.* Tracking results for the PFAR on a test trajectory with w = 10.0 m/s

## G. Proofs of Standard Gradient and Mirror Descent Adaptation

This section presents detailed proofs of the existing methods discussed in the main text, included here for completeness.

Given a tracking controller $\bar{u}$ for the nominal dynamics and its associated stability certificate $\bar{V}(x, r)$, we show in the following lemmas that, under the assumption of linear parameterization, this controller guarantees asymptotic tracking in the presence of disturbances. Building upon the nominal control law $\bar{u}(t)$ and its Lyapunov function $\bar{V}(x, r)$, the adaptive controller is formulated as:

$$
\begin{aligned}
u &= \bar{u} - Y(x)\hat{a}, \\
\dot{\hat{a}} &= \Gamma^{-1} Y(x)^\top g(x)^\top \nabla_x \bar{V}(x, r),
\end{aligned}
\tag{70}
$$

where $\Gamma \in \mathbb{R}^{d \times d}$ is a positive-definite matrix and $\hat{a}$ is the parameter estimate. Using a Lyapunov-like function that accounts for the parameter-estimation error $a - \hat{a}$,

**Lemma G.1.** *Under the adaptive controller defined in* (70)*, consider the Lyapunov-like function*

$$
V(x, r, a, \hat{a}) = \bar{V}(x, r) + \frac{1}{2}(a - \hat{a})^\top \Gamma (a - \hat{a}),
\tag{71}
$$

*where $\Gamma$ is the positive-definite matrix. Then, the time derivative of $V(x, r, a, \hat{a})$ along the closed-loop trajectories satisfies*

$$
\dot{V}(x, r, a, \hat{a}) = \nabla_x \bar{V}(x, r)^\top \big( f(x) + g(x)\bar{u} \big) + \nabla_r \bar{V}(x, r)^\top \big( f(r) + g(r)u(r) \big).
\tag{72}
$$

*Consequently, $V(x, r, a, \hat{a})$ serves as a Lyapunov-like stability certificate for the closed-loop system, and the tracking error converges asymptotically to zero.*

**Proof.** Taking the time derivative of $V(x, r, a, \hat{a})$ along any trajectory $(x(t), r(t), a(t), \hat{a}(t))$, we have

$$
\begin{aligned}
\dot{V} &= \nabla_x \bar{V}(x, r)^T \dot{x} + \nabla_r \bar{V}(x, r)^T \dot{r} + \dot{\hat{a}}^T \Gamma (\hat{a} - a) \\
&= \nabla_x \bar{V}(x, r)^T (f(x) + g(x)(u + Y(x)a)) \\
&\quad + \nabla_r \bar{V}(x, r)^T (f(r) + g(r)u(r)) \\
&\quad + \dot{\hat{a}}^T \Gamma (\hat{a} - a) \\
&= \nabla_x \bar{V}(x, r)^T (f(x) + g(x)\bar{u} - Y(x)(\hat{a} - a)) \\
&\quad + \nabla_r \bar{V}(x, r)^T (f(r) + g(r)u(r)) \\
&\quad + \dot{\hat{a}}^T \Gamma (\hat{a} - a) \\
&= \nabla_x \bar{V}(x, r)^T (f(x) + g(x)\bar{u}) \\
&\quad + \nabla_r \bar{V}(x, r)^T (f(r) + g(r)u(r)) \\
&\quad + \left( \dot{\hat{a}} - \Gamma^{-1} Y(x)^T g(x)^T \nabla_x \bar{V}(x, r) \right)^T \Gamma (\hat{a} - a) \\
&= \nabla_x \bar{V}(x, r)^T (f(x) + g(x)\bar{u}) + \nabla_r \bar{V}(x, r)^T (f(r) + g(r)u(r))
\end{aligned}
\tag{73}
$$

Note that the above time derivative is the same as $\bar{V}$ when $\bar{u}$ is applied to the nominal dynamics $\dot{x} = f(x) + g(x)u$. Using Barbalat's lemma, $V(x, x_r, \hat{a}, a)$ can serve as a certificate that ensures $x \to r$. $\square$

The adaptive controller (70) ensures that the scalar function $V(x, r, a, \hat{a})$ evolves along the trajectories of the disturbed dynamics (1) in the same manner as $\bar{V}(\bar{x}, r)$ does for the nominal system.

Subsequently, a mirror-descent-based controller (Tang et al., 2025) was proposed to guarantee asymptotic tracking in the presence of disturbances.

**Lemma G.2.** *Design the adaptive controller*

$$
\begin{aligned}
u &= \bar{u} - Y(x)\hat{a}, \\
\dot{\hat{a}} &= -P^{-1}(\nabla^2 \chi(P\hat{a}))^{-1} P^{-1} Y(x)^\top g(x)^\top \nabla_x \bar{V}(x, r),
\end{aligned}
\tag{74}
$$

*consider the Lyapunov-like function*

$$
V(x, r, a, \hat{a}) = \bar{V}(x, r) + d_\chi(Pa \| P\hat{a})
\tag{75}
$$

*$P = P^\top \succ 0$, and $\chi : \mathbb{R}^p \to \mathbb{R}$ is a strictly convex potential function. Then, the time derivative of $V(x, r, a, \hat{a})$ along the closed-loop trajectories satisfies*

$$\dot{V}(x, r, a, \hat{a}) = \nabla_x \bar{V}(x, r)^\top \left( f(x) + g(x)\bar{u} \right) + \nabla_r \bar{V}(x, r)^\top \left( f(r) + g(r)u(r) \right). \tag{76}$$

*Consequently, $V(x, r, a, \hat{a})$ serves as a Lyapunov-like stability certificate for the closed-loop system, and the tracking error converges asymptotically to zero.*

**Proof.** First, definite the Bregman divergence,

$$\begin{aligned}
\frac{d}{dt} d_\chi(Pa || P\hat{a}) &= \frac{d}{dt} \left( \chi(Pa) - \chi(P\hat{a}) - \nabla\chi(P\hat{a})^T (Pa - P\hat{a}) \right) \\
&= -\nabla\chi(P\hat{a})^T P\dot{\hat{a}} + (\nabla^2\chi(P\hat{a})P\dot{\hat{a}})^T P\tilde{a} + \nabla\chi(P\hat{a})^T P\dot{\hat{a}} \\
&= (\nabla^2\chi(P\hat{a})P\dot{\hat{a}})^T P\tilde{a}
\end{aligned} \tag{77}$$

where $\tilde{a} = a - \hat{a}$ is the parameter estimation error.

Taking the time derivative of $V(x, r, a, \hat{a})$ along any trajectory $(x(t), r(t), a(t), \hat{a}(t))$, we have

$$\begin{aligned}
\dot{V} &= \nabla_x \bar{V}(x, r)^T \dot{x} + \nabla_r \bar{V}(x, r)^T \dot{r} + \frac{d}{dt} d_\chi(Pa || P\hat{a}) \\
&= \nabla_x \bar{V}(x, r)^T (f(x) + g(x)(u + Y(x)a)) \\
&\quad + \nabla_r \bar{V}(x, r)^T (f(r) + g(r)u(r)) \\
&\quad + (\nabla^2\chi(P\hat{a})P\dot{\hat{a}})^T P\tilde{a} \\
&= \nabla_x \bar{V}(x, r)^T (f(x) + g(x)\bar{u} - Y(x)(\hat{a} - a)) \\
&\quad + \nabla_r \bar{V}(x, r)^T (f(r) + g(r)u(r)) \\
&\quad + (\nabla^2\chi(P\hat{a})P\dot{\hat{a}})^T P\tilde{a} \\
&= \nabla_x \bar{V}(x, r)^T (f(x) + g(x)\bar{u}) \\
&\quad + \nabla_r \bar{V}(x, r)^T (f(r) + g(r)u(r)) \\
&\quad + \left( \dot{\hat{a}} - P^{-1}(\nabla^2\chi(P\hat{a}))^{-1} P^{-1} Y(x)^\top g(x)^\top \nabla_x \bar{V}(x, r) \right)^T P(\hat{a} - a) \\
&= \nabla_x \bar{V}(x, r)^T (f(x) + g(x)\bar{u}) + \nabla_r \bar{V}(x, r)^T (f(r) + g(r)u(r))
\end{aligned} \tag{78}$$

Therefore, the adaptive controller (74) ensures that the scalar function $V(x, r, a, \hat{a})$ evolves along the trajectories of the disturbed dynamics (1) in the same manner as $\bar{V}(\bar{x}, r)$ does for the nominal system. $\qquad\square$

