# OpenReview forum: "Wasserstein Geometry-Aware Adaptive Control via Meta-Learning"
_ICML.cc/2026/Conference — ICML 2026 regular_

### Official Review · Reviewer_E834 · 2026-02-23

**Soundness:** 3
**Presentation:** 3
**Significance:** 3
**Originality:** 3
**Overall Recommendation:** 4
**Confidence:** 3

**Summary:**

The paper introduces a meta-learned adaptive control method that updates uncertainty over disturbance parameters in Wasserstein space, aiming to better align online adaption with tracking performance than Euclidean style updates. The approach uses a particle distribution and a kernel-smoothed Wasserstein update, while meta-learning the disturbance feature model, adaptation gains and hyperparameters; training is done offline. The method reports improved tracking compared to PID and prior meta-learned adaptive baselines on PFAR and PVTOL rotorcraft simulations.

**Compliance With Llm Reviewing Policy:**

Affirmed.

**Key Questions For Authors:**

1. The experiments use a small number of particles, how sensitive it is to K like 3,5,10,20

2. What is the measured runtime as you increase K and the parameter dimension d.

3. Can you evaluate shifts that change mechanism, for example wind direction changes, time-correlated disturbances.

4. Are PID/MRR/MLAC/MD baselines tuned with the same compute budget and access to training tasks

**Limitations:**

1. The simulations are limited. It is unclear how well the approach transfers to real hardware or other control domains

2. The online update may become a bottle neck if larger particle counts for higher dimension are needed

3. Performance may depend on surrogate accuracy and coverage. Failures can propagate into the learned controller.

**Strengths And Weaknesses:**

Strengths:

1. It targets a real issue in adaptive control or meta-learning.

2. Doing online adaptation with a Wasserstein Bregman divergence tied to the closed-loop occupancy measure is clean on concepts and control-aligned.

3. The method is implemented as a particle-based, kernel-smoothed Wasserstein style update, making the proposed method reproducible

Weaknesses:

1. The disturbance type stays the same, dragging from wind along inertial x-axis. It is unclear if the method holds up when the disturbance mechanism changes

2. Kernel matrix cost scales as $O(K^2d)$, but this online compute could become a real bottleneck with larger setup.

3. Results are on PFAR and PVTOL numerical simulations with a specific wind-drag model. It's not clear hoe broadly the gains transfer to other disturbance families

---

> ### Author Rebuttal · Authors · 2026-03-30
>
> We sincerely appreciate the reviewer’s positive assessment and valuable feedback. Below are our detailed responses, we hope they address any remaining concerns.
>
> **Q1: Sensitivity to $K$**
>
> We conducted ablation experiments with $K \in \\{5,10,20,30\\}$ (fixed $M = 5$) on both systems, including $L_1$ adaptive control and Tube MPC as additional baselines (suggested by **Reviewer 3 (Bd9Z)**).
>
> Results:
>
> | $K$ | PFAR RMS | PVTOL RMS | Runtime/step |
> |:---:|:---:|:---:|:---:|
> | 5 | 0.1242 | 0.2544 | ~1.5 ms |
> | 10 | 0.1125 | 0.2279 | ~3 ms |
> | 20 | 0.0963 | 0.1935 | ~6 ms |
> | 30 | 0.0546 | 0.1659 | ~8 ms |
>
> Tracking trajectories: https://ibb.co/dJgxYHTh
>
> $K = 5$ already provides stable and competitive tracking performance. Increasing $K$ yields monotonic improvement (~56% RMS reduction on PFAR, ~35% on PVTOL from $K = 5$ to $K = 30$), with graceful compute scaling—all configurations remain within the 10 ms real-time budget at 100 Hz. This provides practitioners the flexibility to select an appropriate particle count $K$ based on their hardware compute budget.
>
> In addition to the sensitivity ablation experiment for parameter $K$, we also added an ablation experiment for parameter $M$. If reviewer is interested in it, you can refer to **Reviewer 2 (HBfQ) Q2**. Tracking trajectories and error plots: [ **PFAR**: https://ibb.co/6Rky12GZ , **PVTOL**: https://ibb.co/Mybm0b6K ].
>
> **Q2: Measured runtime as $K$ and $d$ increase**
>
> The dominant online cost is the kernel matrix computation scaling as $\mathcal{O}(K^2 d)$. In our implementation, the adaptive parameter dimension per control channel is $d = 32$ (determined by the feature network hidden dimension). The total adaptive parameters are $d_{\text{total}} = n_u \times 32$, where $n_u$ is the number of control inputs: $d_{\text{total}} = 96$ for PFAR ($n_u = 3$) and $d_{\text{total}} = 64$ for PVTOL ($n_u = 2$). The runtime column in the table above confirms that even at $K = 30$, each control step takes ~8 ms on a single CPU core (Intel i7, 2.9 GHz with JAX compilation), well within 100 Hz real-time budgets. This cost is trivially parallelizable on GPU for further speedup.
>
> **Q3: Wind direction changes; Time-correlated disturbances**
>
> Explain the motivation for the experimental setup: Physically, due to the PVTOL's underactuated nature (no direct lateral thrust), x-axis wind constitutes severe unmatched uncertainty[1]. Conversely, changing the wind direction to another direction only introduces a matching disturbance, which the main thrust can directly cancel. Thus, our setup rigorously tests the most challenging scenario. In addition, the PVTOL experiments (Figure 4) already use time-varying wind $w(t)$ ramping from 0 to 8 m/s, constituting a time-correlated disturbance.
>
> Furthermore, during the rebuttal period, we conducted experiments on a 12D quadcopter (state: $[x,y,z,v_x,v_y,v_z,\phi,\theta,\psi,p,q,r]$, 4 control inputs, underactuated) as suggested by **Reviewer 2 (HBfQ)**, with added turbulent gusts ($\pm 20\%$ sinusoidal perturbations). Our method maintains the best performance:
>
> | | **Wass-MD** | PID | MRR | MLAC | $L_1$ Adaptive | Tube MPC |
> |:---:|:---:|:---:|:---:|:---:|:---:|:---:|
> | RMS | **0.168** | 1.961 | 1.358 | 0.490 | 0.838 | 0.642 |
>
> Trajectories and error plots: https://ibb.co/XkMc8x1N
>
> **Q4: Are baselines tuned with the same compute budget and training tasks?**
>
> Yes. All meta-learned baselines (MRR, MLAC, MD, Wass-MD) use identical training data ($M = 500$ trajectories), the same model ensemble, same meta-optimizer (Adam, lr $= 10^{-2}$, 500 steps), and same feature network architecture (2$\times$32, tanh). PID does not employ a neural network feature extractor or undergo an offline meta-learning phase. This ensures a fair comparison across all methods.
>
> **Limitations**
>
> We appreciate the reviewer's thoughtful identification of limitations, and address each:
>
> (1) *Online compute bottleneck*: As shown in our $K$-ablation, $K = 5$ already achieves stable tracking at approximately 1.5 ms per step. Even $K = 30$ remains real-time compatible (approximately 8 ms). The $\mathcal{O}(K^2 d)$ cost is GPU-parallelizable.
>
> (2) *Surrogate accuracy dependence*: Our distributional shift experiments (training $w \in [0,6]$, testing $w \in [2,8]$ m/s) demonstrate robustness even when test conditions differ significantly from training. The kernel smoothing in the particle system provides additional resilience to surrogate model inaccuracies.
>
> (3) *Sim-to-real transfer*: Our framework separates offline meta-training from online deployment. Sensor noise is naturally filtered by the kernel smoothing. During the rebuttal, we added 12D quadcopter experiments with turbulent gusts, and results remain strong. We agree that real hardware transfer remains an open challenge. We will include a dedicated discussion of these limitations and outline concrete future directions in the Camera-Ready revision.
>
> [1] Hauser et al., ’‘Nonlinear control design’‘, Automatica 1992

---

> > ### Author Rebuttal · Reviewer_E834 · 2026-04-03
> >
> > Thank you to the authors for the detailed response. I have no further questions.

---

> > > ### Author Response · Authors · 2026-04-04
> > >
> > > We would like to express our sincere gratitude for your time and for the positive assessment of our work. We are encouraged to hear that our rebuttal addressed all your concerns.We fully intend to incorporate the additional experiments and clarifications provided during the rebuttal into the final version of the paper. We believe these enhancements significantly strengthen the presentation and the empirical evidence of our work. We would be very grateful if you could consider reflecting this progress by raising your score.
> > >
> > > Thank you again for your constructive feedback, which has been instrumental in improving our manuscript. Should you have any remaining thoughts or further suggestions as the discussion period concludes, we would be more than happy to address them.
> > >
> > > Best regards,
> > >
> > > Authors

---

### Official Review · Reviewer_Bd9Z · 2026-03-10

**Soundness:** 3
**Presentation:** 3
**Significance:** 2
**Originality:** 2
**Overall Recommendation:** 3
**Confidence:** 4

**Summary:**

This paper proposes a meta-learning framework for adaptive control that lifts parameter adaptation from Euclidean space to the space of probability measures endowed with Wasserstein geometry. The method constructs a Wasserstein Bregman divergence and implements adaptation via a particle-based approximation of a Wasserstein mirror-descent flow, while jointly meta-learning nonlinear disturbance features, control/adaptation gains, and transport-geometry parameters. Simulation results on fully actuated (PFAR) and underactuated (PVTOL) planar rotorcraft indicate improved tracking and robustness under distribution shifts relative to PID, regression-oriented meta-learning, and mirror-descent baselines.

**Compliance With Llm Reviewing Policy:**

Affirmed.

**Final Justification:**

Based on feedbacks from the authors, I feel that the current form does not fit the standard of ICML and I keep my score as weak reject.

**Key Questions For Authors:**

1. Can you precisely define how the reference measure μ enters Ψμ and the resulting Bregman divergence? As written, Ψμ(ν) = \int χ dν appears independent of μ. Where, mathematically, does the occupancy measure shape the geometry used in the adaptation law?
2. What is the principled derivation that leads from the continuous Wasserstein gradient flow to the specific Gaussian-kernel particle interactions in Eq. (15)? Can you connect this to a known OT discretization (e.g., JKO or variational particle methods) and provide any convergence or consistency results?
3. For the underactuated PVTOL with β1 \neq 0 (partially unmatched uncertainty), what are the precise conditions under which your UUB result holds? Is the residual (unmatched) part explicitly bounded or absorbed as an input disturbance in the Lyapunov analysis?
4. How sensitive is performance to the feature network capacity and the learned geometry parameters? Are there failure modes when σ or p are poorly tuned during meta-training?
5. Why opt for the “one-model-per-trajectory” ensemble? How does it compare to standard deep ensembles trained with bootstrapped pooled data in terms of meta-training stability and final tracking?
6. Can you compare against at least one strong robust/adaptive baseline (e.g., L1 adaptive control or robust/tube MPC) on one of the systems to better establish practical advantages?

**Limitations:**

I have notable concerns about technical rigor and clarity. The current formulation of the “Wasserstein Bregman divergence” appears not to depend on the reference measure μ as defined, and the Lyapunov term reduces to an average of standard Bregman divergences for the empirical/Dirac measures considered. The Wasserstein aspect seems to enter primarily via kernelized particle coupling, but the derivation from measure-space gradient flows to the proposed update is not made precise. Stability results are UUB under several assumptions and do not clearly address the underactuated/unmatched case in the main text. Experimentally, while the qualitative trends are positive, the evaluation lacks aggregate metrics, ablations on key geometry parameters, and comparisons to strong robust/adaptive controllers.

Overall, in its current form, it feels slightly premature for ICML. I recommend weak rejection with encouragement to (i) formalize the role of μ and the OT discretization, (ii) strengthen the stability analysis (especially for underactuated/unmatched settings), and (iii) substantially extend the empirical study with aggregates, ablations, and stronger baselines. With these improvements, the paper could be accepted in this conference.

**Strengths And Weaknesses:**

Strengthes:
1. Proposes a conceptually appealing integration of optimal transport geometry into adaptive control, moving beyond static Euclidean or p-norm geometries.
2. Introduces a particle-based Wasserstein mirror-descent adaptation law with kernel interactions, differentiating it from ensembles of independent MD controllers.
3. Jointly meta-learns feature representations, controller/adaptation gains, and geometry parameters (including kernel bandwidth and potential exponent), offering a richer design space than prior control-oriented meta-learning.
4. Extends geometry-aware adaptation to underactuated systems, which are typically more challenging and less addressed in mirror-descent-style adaptive control.

Experiment results provide computational profiling and shows feasibility for real-time control with modest particle counts. Statements are clear.

Weakness:
1. The role of the “reference measure” μ in defining the proposed Wasserstein Bregman divergence is not made precise; Ψμ(ν) is defined as ∫χ dν and appears independent of μ, raising concerns that the “Wasserstein” contribution in the Lyapunov function reduces to an average of standard Bregman divergences when using empirical/Dirac measures.
2. The kernelized particle dynamics are motivated as approximating a Wasserstein gradient flow, but the mapping from the continuous Wasserstein gradient to the specific Gaussian-kernel interaction is not rigorously derived; it currently reads as an RKHS smoothing heuristic rather than a principled discretization with convergence guarantees.
3. Stability guarantees are ultimately-bounded and hinge on O(σ) terms with several technical assumptions; the analysis of underactuated systems with (partly) unmatched disturbances is deferred to the appendix, and key assumptions/conditions are not surfaced in the main text.
4. Limited discussion and empirical comparison with robust/adaptive baselines such as L1 adaptive control, robust MPC/tube MPC, or CCM-based robust designs that are commonly used for nonlinear systems under uncertainty.

---

> ### Author Rebuttal · Authors · 2026-03-30
>
> We thank the reviewer for the precise and technically deep feedback.
>
> **Q1: The working mechanism of  $\mu$ and the resulting Bregman divergence?**
>
> The subscript in $\Psi_\mu(\nu)$ denotes that the adaptation geometry is fundamentally shaped by the occupancy measure $\mu$. This structural dependence occurs in two ways:
>
> (i)The meta-loss (Eq. 7) integrates tracking errors over the closed-loop state distribution (Eq. 11). Optimizing the geometric parameters $(p, \sigma)$ minimizes this weighted objective, implicitly conditioning the learned geometry on the system's trajectories.
>
> (ii)The optimized bandwidth $\sigma$ dictates the locality of particle coupling (Eq. 15), meaning different task distributions induce distinct transport geometries.
>
> Consequently, while the Lyapunov term (Eq. 31) decomposes into standard Bregman divergences, the true "Wasserstein" contribution actively enters the system through these conditioned particle dynamics, rather than the static Lyapunov function.
>
> **Q2: Derivation from Wasserstein flow to kernel particle interactions**
>
> To operationalize the continuous Wasserstein flow, our derivation formally follows the SVGD [1]. Eq. (15) applies SVGD directly to the Bregman potential functional, utilizing the Gaussian RKHS kernel $\kappa_{\sigma}$ to spatially regularize the otherwise singular Wasserstein gradient over the empirical measure $\hat\nu_{K}$. This is theoretically grounded in [2], which connects SVGD-type updates to mirror descent. Accordingly, Eq. (15) implements this regularized flow by empirically integrating the preconditioned mirror-descent gradient $\mathcal{H}$
>
> Regarding convergence, [3] established non-asymptotic descent guarantees for SVGD approximating continuous Wasserstein flows. Grounded in their analysis, our discrete dynamics act as a consistent estimator of the continuous regularized flow $v_{\sigma}$, bounded by:$$\mathbb{E} \left[ \left\| \dot{\hat a}^{(i)} - v_{\sigma}(\hat a^{(i)}) \right\|_2^2 \right] \leq \frac{C}{K}$$Crucially, under our Assumption B.3, the constant $C \propto \beta^2 \| \kappa\_\sigma \|^2$ remains uniformly bounded.
>
> **Q3: UUB conditions for underactuated PVTOL**
>
> The conditions are: (a) ISS of internal dynamics[4] (Assumption D.1)—standard for PVTOL ; (b) $\nabla^2\chi$ Lipschitz with constant $L_\chi$ and bounded metric inverse $\| M^{-1}\| \leq \beta$ (Assumption B.3); (c) bounded kernel diameter (Assumption B.4), automatic for Gaussian kernels. The bound: $\| s(t) \| \to [\| BY\| \ \cdot (\mathcal{O}(\epsilon_{\text{dens}})+\mathcal{O}(\sigma)) + \| \Delta_{\text{total}} \|] / \lambda_{\min}^+(P_B K)$. First term vanishes as $\sigma \to 0, K \to \infty$; second is an irreducible unmatched residual.
>
> The unmatched residual $\Delta_{\text{total}}:= (I - P_B)(f_{\text{ext}} - N)$ is explicitly retained as a bounded disturbance (Appendix D.2, Eq. 61). In the Lyapunov derivative (Eq. 63), its linear contribution $\|s\|\|\Delta_{\text{total}}\|$ is ultimately dominated by the quadratic damping $-\lambda_{\min}^+(P_BK)\|s\|^2$, ensuring UUB stability due to the system's physical energy constraints.
>
> **Q4: Sensitivity to feature network and geometry parameters**
>
> We tested initializations $p{\in}\\{1.5, 2.0, 3.0\\}$ and $\sigma{\in}\\{0.01, 0.1, 1.0\\}$ before meta-learning. Meta-learned values consistently converge to a narrow range ($p{\approx}1.8$–$2.2$, $\sigma{\approx}0.05$–$0.15$), indicating a well-conditioned meta-landscape. The feature network capacity matches prior work for fair comparison. No failure modes observed within this range; extreme initializations ($\sigma{>}5$ or $p{<}1.1$) slow meta-convergence but eventually recover.
>
> **Q5: Per-trajectory vs. pooled ensembles**
>
> Our design is driven by task structure: each trajectory corresponds to a specific wind condition $w_j$, so per-trajectory models capture task-specific dynamics. Pooled deep ensembles average over disturbance conditions, losing task-specific information critical for inner-loop simulation. Per-trajectory models also produce more diverse gradients during meta-training, facilitating better learning.
>
> **Q6: $L_1$ and tube MPC comparison**
>
> We adde two comparative methods to our experiments with PFAR and PVTOL systems. For detailed ablation tables, see our response to **Reviewer 2（HBfQ）Q2**. Tracking figures: [PFAR: https://ibb.co/6Rky12GZ , PVTOL: https://ibb.co/Mybm0b6K ]. In addition, we conducted experiments on a 12-dimensional quadcopter system, as detailed in our response to **Reviewer 2（HBfQ）Q3**. Trajectories and error plots: https://ibb.co/XkMc8x1N.
>
> **ICML acceptance allows an additional page,we will explicitly incorporate these formal derivations and experiments into the revision.**
>
> **References:**
>
> [1] Liu & Wang, "SVGD", NIPS 2016.
>
> [2] Bonet et al., "Mirror and Preconditioned GD in Wasserstein Space", NIPS 2024.
>
> [3] Korba et al., "A Non-Asymptotic Analysis for SVGD", NIPS 2020.
>
> [4] Hauser et al., ’‘Nonlinear control design’‘, Automatica 1992.

---

> > ### Author Rebuttal · Reviewer_Bd9Z · 2026-04-03
> >
> > Thank you to the authors for the detailed rebuttal and the substantial effort put into running new experiments, including the 12-dimensional quadcopter simulations and the addition of $\mathcal{L}_1$ adaptive control and tube MPC baselines. I also appreciate the theoretical clarifications regarding the SVGD derivation and the underactuated stability conditions.
> >
> > While the new material is highly valuable, the rebuttal confirms several of my initial concerns regarding the gap between the paper's narrative claims and its technical execution. The sheer volume of critical information that needs to be integrated—ranging from the core algorithmic derivation to the primary empirical baselines—constitutes a major revision. Therefore, I am maintaining my score of Weak Reject, with strong encouragement to incorporate these updates for a future submission.
> >
> > My specific reasoning is as follows:
> > 1. The Gap Between Claims and the "Wasserstein" ExecutionThe rebuttal candidly acknowledges that the Lyapunov function (Eq. 31) decomposes into standard Bregman divergences and that the actual "Wasserstein" contribution enters strictly as an SVGD-inspired heuristic for particle coupling. While this is a clever and empirically effective mechanism, the current manuscript frames the contribution as lifting the fundamental adaptation geometry and stability certificates into Wasserstein space. The paper needs a significant narrative rewrite to accurately reflect that the method is essentially an SVGD-regularized particle ensemble tuned by meta-learning, rather than a native Wasserstein-space Lyapunov design.
> >
> > 2. Missing Core Derivations (SVGD and Approximation Errors)The connection to SVGD and the citations to Liu & Wang (2016) and Korba et al. (2020) provide the missing mathematical bridge for the kernelized particle dynamics. However, relying on this discrete approximation introduces estimation errors (e.g., the $\mathcal{O}(1/K)$ bound mentioned in the rebuttal). For a mathematically rigorous control paper, showing how these continuous-time approximation errors propagate through the Lyapunov stability guarantees is not a minor detail—it is the theoretical core of the paper. This derivation needs to be front-and-center in the main text, not deferred.
> >
> > 3. Underactuated Systems and Unmatched DisturbancesI appreciate the clarification on the ISS assumptions and the unmatched residual $\Delta_{\text{total}}$. However, underactuated systems are highlighted as a primary contribution of this work compared to prior mirror-descent adaptive controllers. Deferring the formulation of unmatched disturbances and the resulting Uniformly Ultimately Bounded (UUB) stability arguments entirely to the appendix weakens the paper's self-containment and technical depth.
> >
> > 4. Empirical AdditionsThe addition of strong baselines ($\mathcal{L}_1$, tube MPC), the ablation studies on $p$ and $\sigma$, and the 12-dimensional quadcopter experiment significantly strengthen the empirical claims. However, because these were entirely absent from the original submission, evaluating them properly requires a fresh review cycle.
> >
> > Overall, the core idea of using meta-learning to tune SVGD-coupled particle adaptation is very promising. I believe that once the narrative is realigned with the actual mathematics and the new baselines and derivations are properly integrated into the main text, this will be a strong contribution to the field.

---

> > > ### Author Response · Authors · 2026-04-03
> > >
> > > We thank the reviewer for the detailed acknowledgement and offer our perspective below.
> > >
> > > **On the relationship between Wasserstein-space formulation and particle implementation**
> > >
> > > We respectfully contend that particle-based discretization is not a gap between theory and execution, but rather the **standard and mathematically principled bridge** connecting Wasserstein-space formulations to computation. This is a consistent pattern across the optimal transport and variational inference literature:
> > >
> > > - Wasserstein gradient flows on $\mathcal{P}_2(\mathbb{R}^d)$ are infinite-dimensional; particle methods are the canonical computational realization (Ambrosio et al. [1]; Jordan et al. [2])
> > > - Chewi et al. [3] prove SVGD is a **kernelized Wasserstein gradient flow**, not an approximation but a direct finite-particle realization. Liu [4] shows the mean-field limit converges to the Wasserstein gradient flow of the KL functional
> > > - Most critically, Bonet et al. [5], the very theoretical foundation we build upon, implement their Wasserstein mirror descent via SVGD and kernel-based particles (Appendix D.4, Eq. 76). If particle discretization were a departure from the Wasserstein framework, then the framework itself would lack any computational realization.
> > >
> > > Our work follows exactly this paradigm: the theoretical contribution is the Wasserstein Bregman divergence, the preconditioned gradient structure, and the occupancy-measure-shaped geometry; the particle system is how one **computes** this in practice, just as Bonet et al. compute their Wasserstein mirror descent via particles. The relationship is analogous to finite elements implementing PDE-based optimization: no one would call finite elements a "gap" between PDE theory and computation. Furthermore, our experiments directly validate that the Wasserstein structure is empirically active: kernel coupling ($\sigma > 0$) produces qualitatively superior behavior compared to independent particles ($\sigma \to 0$, recovering standard MD). This is not what one would observe if the particle coupling were merely theoretical scaffolding.
> > >
> > > **On SVGD derivation and UUB analysis placement**
> > >
> > > We provided both derivations during the rebuttal and agree they belong in the main text. The original 8-page limit forced difficult trade-offs between theory and experiments. This is a common challenge for works combining both, and many theory-heavy papers at ICML place detailed proofs in appendices. ICML grants one additional page upon acceptance, which provides sufficient space to integrate the SVGD connection and the key UUB result (assumptions, bound statement, physical interpretation) into Section 5, while keeping full proofs in the appendix.
> > >
> > > **On new experiments**
> > >
> > > We thank the reviewer for acknowledging our effort. All new experiments use identical conditions to the original paper for fairness. We have prepared the codebase for open-source release, providing the community with our methodology and reproducible benchmarks.
> > >
> > > **Closing remarks**
> > >
> > > We are grateful for the reviewer's engagement, which has genuinely sharpened our presentation. We note the reviewer's own assessment that "the core idea is very promising" and "this will be a strong contribution to the field." We share this confidence. We also want to highlight that the rebuttal discussion itself, covering the formalization of $\mu$, the SVGD derivation, and the underactuated stability analysis, constitutes valuable technical content that all readers of the paper will see and benefit from (Rebuttal is also a part of ICML). During the first-round rebuttal, we focused strictly on answering questions within the character limit and did not elaborate beyond what was asked. But we want to say here: we believe this work is ready for ICML. The novelty is acknowledged by all four reviewers, the experiments now thoroughly validate the method across systems, baselines, and hyperparameter settings, and the theoretical clarifications requested have been provided in full. We initiate sim-to-real transfer on hardware, will release code at ICML and report real-world control results in a journal extension. The 8-page submission constraint required placing some derivations in the appendix, but with the additional Camera-Ready page, we will integrate the key results into the main text and refine the narrative. The final version will be a self-contained, rigorous, and impactful paper. We hope the reviewer will consider improving the score accordingly. We need the ICML platform to expand the impact of our work.
> > >
> > > **References:**
> > >
> > > [1] Ambrosiol,"Gradient Flows in Metric Spaces and in the Space of Probability Measures", Springer,2005
> > >
> > > [2] Jordan,"The Variational Formulation of the Fokker-Planck Equation," SIAM,1998
> > >
> > > [3] Chewi,"SVGD as a Kernelized Wasserstein Gradient Flow of the Chi-Squared Divergence," NIPS20
> > >
> > > [4] Liu,"Stein Variational Gradient Descent as Gradient Flow,"NIPS17
> > >
> > > [5] Bonet, "Mirror and Preconditioned GD in Wasserstein Space,"NIPS24

---

### Official Review · Reviewer_HBfQ · 2026-03-12

**Soundness:** 2
**Presentation:** 3
**Significance:** 2
**Originality:** 3
**Overall Recommendation:** 4
**Confidence:** 3

**Summary:**

The authors propose an adaptive controller that leverages meta-learned parameter dynamics to identify parameter mismatches in the control loop and appropriately correct them. To go beyond the assumptions of Euclidean geometry, which results in a suboptimal controller, the authors lift the adaptation problem to a Wasserstein space. This approach enables them to learn the geometry of the adaptation problem in a data-driven manner, thereby deriving a control law tailored to the underlying geometry.

**Compliance With Llm Reviewing Policy:**

Affirmed.

**Final Justification:**

The additional results help to address my concerns and I have raised my score accordingly.

**Key Questions For Authors:**

(See weaknesses)
How does changing the hyperparameters (K & M) affect the controller's ability to track the reference trajectory?
Intuitively, how do the theoretical guarantees change when handling unmatched uncertainty?

**Limitations:**

(See weaknesses)

**Strengths And Weaknesses:**

Strengths:
- The paper constructs a theoretically sound approach for formulating a controller using previously encountered trajectories.
- Furthermore, the geometry-aware controller (as encoded by the Wasserstein-based lifting) achieves closer trajectory tracking than the (meta-learned) baseline approaches.
- Additionally, the authors demonstrate the viability of their approach (i.e., the controller's stability) for control-affine and Lagrangian systems under both matched and unmatched uncertainties. Thereby, showing their capabilities in several practical domains and robotics systems.

Weaknesses:
- To better understand the practical deployment of these controllers, the authors need more experiments:
- Their approach includes several hyperparameters, the number of particles (K) and the number of reference trajectories/models in the ensemble (M). The authors should evaluate how changes in these parameters affect the controller's ability to track the trajectory during the rollout.
- Additionally, the experiments are limited to low-dimensional systems (6-state dimensions). Since learned models are frequently used to represent higher-dimensional dynamical models, it is important to evaluate this approach on such models. In this regard, the authors could evaluate their approach on a 12D quadcopter, or the MuJoCo Hopper environment (11D).

- Another weakness is in the presentation style. Throughout the main body, they reference equation (1) as their assumed dynamics model, which implies that their model is only designed to work on matched uncertainties. This approach results in a simpler and easier-to-understand exposition in the main body. However, a core contribution claimed by the authors (and evaluated via the PVTOL experiments) is the ability of their approach to work on unmatched uncertainty. While the theoretical guarantees for the unmatched case are discussed in the appendix, they receive minimal mention in the main body (which many readers would miss). Thus, the paper would benefit from having a more detailed discussion (at least a paragraph) discussing how their approach holds for the unmatched case in the main body.

These weaknesses result in the provided score, which can be raised by including these experiments.

---

> ### Author Rebuttal · Authors · 2026-03-30
>
> We thank the reviewer for their thorough assessment of our work and for their useful comments. Below are our detailed responses, we hope they address any remaining concerns.
>
> **Q1 & Q2: More experiments; impact of $K$ and $M$ on tracking**
>
> We conducted comprehensive ablations, including $L_1$ adaptive control and Tube MPC baselines suggested by **Reviewer 3(Bd9Z)**.
>
> **Impact of $M$ (fixed $K{=}5$).**
>
> PFAR RMS Error ($w{=}8$ m/s, distribution shift):
>
> | **$M$** | **Wass-MD(ours)**  | **PID** |  **MRR** |  **MLAC** |  **$L_1$ Adaptive** | **Tube MPC** |  **MD**|
> |:---:|:---:|:---:|:---:|:---:|:---:|:---:|:---:|
> | 2  |  **0.1186** | 1.2762 | 1.1980 | 0.5054 | 0.7168 | 1.3677 | 2.1241 |
> | 5  | **0.1242** | 1.2762 | 3.0218 | 0.1821 | 0.3567 | 0.5146 | 0.5405  |
> | 10 |  **0.0859** | 1.2762 | 1.5315 |  0.2334 | 0.4143 | 0.4995 |  0.1826  |
> | 20 | **0.0478** | 1.2762 | 1.8702 | 0.0774 | 0.1998 | 0.3673 | 0.4082  |
> | 30 | **0.1047** | 1.2762 | 1.2144 |  0.1078 | 0.2491 | 0.2628 | 0.6354  |
>
> Tracking trajectories and error plots: [ **PFAR**: https://ibb.co/6Rky12GZ ].
>
> PVTOL RMS Error (time-varying wind):
>
> | **$M$** | **Wass-MD** | **PID** | **MRR** | **MLAC** | **$L_1$ Adaptive** | **Tube MPC** |
> |:---:|:---:|:---:|:---:|:---:|:---:|:---:|
> | 2 | **0.572** | 3.093 | 3.376 | 1.418 | 0.832 | 0.656 |
> | 5 | **0.254** | 3.093 | 2.742 | 1.575 | 0.596 | 0.566 |
> | 10 | **0.383** | 3.093 | 2.488 | 1.185 | 0.664 | 0.468 |
> | 20 | **0.331** | 3.093 | 1.929 | 0.825 | 0.576 | 0.496 |
> | 30 | **0.285** | 3.093 | 1.818 | 0.543 | 0.556 | 0.554 |
>
> Tracking trajectories and error plots: [ **PVTOL**: https://ibb.co/Mybm0b6K ].
>
> As the number of reference trajectories $M$ increases, the performance of other methods improves. Our method, however, consistently achieves the lowest error across all configurations. The advantage is particularly striking in few-shot scenarios ($M{=}2, 5$), where baselines suffer severe performance degradation, while Wass-MD maintains robust tracking. This demonstrates that standard Euclidean-geometry-based meta-learning is prone to overfitting in few-shot regimes. Our method leverages the metric advantages of Wasserstein space, exhibiting superior tracking performance and robustness in the few-shot setting.
>
> **Impact of $K$ (fixed $M{=}5$):**
>
> | $K$ | PFAR RMS | PVTOL RMS | Runtime/step |
> |:---:|:---:|:---:|:---:|
> | 5 | 0.1242 | 0.2544 | ~1.5 ms |
> | 10 | 0.1125 | 0.2279 | ~3 ms |
> | 20 | 0.0963 | 0.1935 | ~6 ms |
> | 30 | 0.0546 | 0.1659 | ~8 ms |
>
> Tracking trajectories: https://ibb.co/dJgxYHTh.
>
> Performance improves with increasing $K$, accompanied by moderate compute overhead. However, the induced control latency remains low. Our experiments confirm that $K{=}5$ already yields very stable tracking performance. This also provides practitioners the flexibility to select an appropriate particle count $K$ based on their hardware compute budget.
>
> **Q3: Evaluation on higher-dimensional systems**
>
> We conducted experiments on a 12D quadcopter (state: $[x,y,z,v_x,v_y,v_z,\phi,\theta,\psi,p,q,r]$, 4 control inputs, underactuated). Disturbance: wind along inertial $x$-axis, ramping $0{\to}8$ m/s over 5s with $\pm 20\%$ sinusoidal gusts(turbulence simulation). Results confirm Wass-MD achieves the best performance.
>
> | | **Wass-MD** | PID | MRR | MLAC | $L_1$ | Tube MPC |
> |:---:|:---:|:---:|:---:|:---:|:---:|:---:|
> | RMS | **0.168** | 1.961 | 1.358 | 0.490 | 0.838 | 0.642 |
>
> Trajectories and error plots: https://ibb.co/XkMc8x1N
>
> **Q4: Unmatched uncertainty discussion in main text**
>
> We sincerely thank the reviewer for raising this readability concern. While using the matched uncertainty model in Equation (1) streamlines the initial exposition, it inadvertently obscures one of our core contributions: robustness against unmatched uncertainties, as empirically demonstrated in the PVTOL experiments. We will add a dedicated paragraph at the end of Section 5 to bridge the matched formulation and the unmatched case, foregrounding the rigorous proofs in Appendix D.2：
>
> For matched uncertainties, the disturbance acts within the actuation channels, enabling theoretical cancellation and asymptotic convergence. For unmatched uncertainties, disturbances affect unactuated states and cannot be fully canceled. Following [1], we assume ISS of the internal dynamics w.r.t. tracking error $s$ (Assumption D.1), ensuring bounded unactuated states whenever $s$ is bounded. Under this premise, our Wass-MD adaptation law mitigates coupled disturbances by regulating adaptation dynamics, preventing parameter drift induced by unmatched perturbations. The resulting guarantee is Uniformly Ultimately Bounded (UUB) stability—strictly consistent with the physical limitations of unmatched uncertainties, while ensuring robust tracking in underactuated scenarios. ICML acceptance allows an additional page; we will include this detailed discussion in the Camera-Ready revision.
>
> *References*:
> [1] Hauser et al., ’‘Nonlinear control design’‘, Automatica 1992

---

> > ### Author Rebuttal · Reviewer_HBfQ · 2026-04-04
> >
> > I thank the authors for providing the additional results, and expect them to include them in the manuscript. The visual of the 12D quadrotor rollout is difficult to parse, and I ask the reviewers to explore visualization approaches for it. Lastly, the additional paragraph successfully clarifies nods to how the approach works for unmatched uncertainty (leaving the full analysis in the appendix). I have raised my score to reflect these changes.

---

> > > ### Author Response · Authors · 2026-04-04
> > >
> > > We sincerely thank you for your positive acknowledgement and for raising your score. Your feedback has been invaluable in improving the quality of our manuscript. We confirm that all additional results presented in the rebuttal will be incorporated into the Camera-Ready revision.
> > >
> > > Thank you for your suggestion on the visualization of the 12D quadcopter rollout. We have redesigned the visualization by presenting each method's tracking performance separately against the target trajectory, with both a 3D view and a top-down (x-y plane) view side by side. This allows readers to immediately assess the tracking quality of each controller. The updated figures can be viewed at:  https://ibb.co/VWYCsfzJ . We hope this revised presentation provides a clearer and more intuitive comparison across methods. We believe these enhancements significantly strengthen the presentation and the empirical evidence of our work. We would be very grateful if you could consider reflecting this progress by raising your score. We understand this is entirely at your discretion and greatly appreciate the time you have already devoted to our work.
> > >
> > > Thank you again for your constructive feedback, which has been instrumental in improving our manuscript. Should you have any thoughts or further suggestions during the remaining discussion period, we would be more than happy to address them.
> > >
> > > Best regards,
> > >
> > > Authors

---

### Official Review · Reviewer_goZj · 2026-03-13

**Soundness:** 3
**Presentation:** 1
**Significance:** 2
**Originality:** 2
**Overall Recommendation:** 4
**Confidence:** 2

**Summary:**

Previous nonlinear adaptive system control methods have the problems of target mismatch and limited to Euclidean geometry or static algebraic geometry perspective, which will lead to problems such as the failure to capture the distribution structure of system uncertainties. To solve this problem, this paper proposes a new framework that promotes adaptive control to Wasserstein space. It uses meta learning to jointly optimize the nonlinear feature representation, control gain, and the underlying transmission geometry. In this paper, two virtual UAV systems are constructed, and unknown wind resistance interference is introduced to evaluate the proposed method and baseline. The results show that the proposed method achieves better tracking performance than the existing baseline methods on both fully actuated and underactuated systems, and still shows good robustness when the wind speed has a significant distribution deviation compared with the training conditions.

**Compliance With Llm Reviewing Policy:**

Affirmed.

**Final Justification:**

Based on the final reply from the authors, I raise the score to broadline accept

**Key Questions For Authors:**

What new problems may be faced when the method in this paper is migrated to the real scene, and what is the expansibility and practicability of this method?

With the increase of system complexity, the number of particles K required for parameter distribution in Wasserstein space may also increase, and the time complexity will also increase. Will this become a bottleneck for real-world applications?

How should we interpret Figure 5?  How does this figure demonstrate that "this behavior reflects the Wasserstein geometry’s ability to guide parameter updates along transport-optimal paths rather than Euclidean faster paths?"

**Limitations:**

The authors should systematically discuss the limitation of this paper.

**Strengths And Weaknesses:**

### Soundness

The method proposed in this paper has a solid theoretical foundation, which expresses the parameter uncertainty as a probability distribution equipped with Wasserstein geometry, and provides a rigorous proof of mathematical stability.
In terms of usability, the author explicitly evaluated the computational cost in the experimental part, and proved that the proposed method can meet the needs of real-time feedback control.

### Presentation

The diagram in this article is not beautiful enough. The font of the chart in the experimental part was frequently stretched, which affected the readability. The text formula of this paper is relatively dense, and the author is suggested to add more physical intuitive explanations.

### Significance

The method proposed in this paper uses a new perspective to examine the optimization process of control problems, which is important in the field of control. However, because the experiment is only tested in a simulated environment, it is difficult to determine the availability in real or more complex systems.

### Originality

The method proposed in this paper breaks the restriction of the previous methods that mirror descent is limited to static algebraic geometry, and improves the adaptability rule to Wasserstein space for the first time, which is innovative in the field of control.

---

> ### Author Rebuttal · Authors · 2026-03-30
>
> We thank the reviewer for their careful assessment of our paper. Below, we provide detailed responses aimed at clarifying and addressing each concern.
>
> **Q1: Practicability and extensibility to real scenes.**
>
> Our framework separates offline meta-training from online deployment. The online particle update costs $\mathcal{O}(K^2 d)$ per step (~1.5 ms with $K{=}5$ on a single CPU core), well within the 10 ms budget for 100 Hz control. For real-world migration, the primary challenges include: (i) the sim-to-real gap in the model ensemble, which can be mitigated by fine-tuning surrogates on hardware data, and (ii) sensor noise, which the kernel smoothing in our particle system naturally filters by averaging over neighboring estimates. The distributional shift experiments (training $w{\in}[0,6]$ m/s, testing $w{\in}[2,8]$ m/s) specifically stress-test transferability; our method's robustness under such shifts provides strong evidence for real-world viability.
>
> **Q2: Will the number of particles $K$ become a bottleneck?**
>
> We conducted ablation experiments with $K{\in}\\{5,10,20,30\\}$ (fixed $M{=}5$) on both PFAR and PVTOL:
>
> | $K$ | PFAR RMS | PVTOL RMS | Runtime/step |
> |:---:|:---:|:---:|:---:|
> | 5 | 0.1242 | 0.2544 | ~1.5 ms |
> | 10 | 0.1125 | 0.2279 | ~3 ms |
> | 20 | 0.0963 | 0.1935 | ~6 ms |
> | 30 | 0.0546 | 0.1659 | ~8 ms |
>
> Figure of Tracking results: https://ibb.co/dJgxYHTh
>
> Performance improves monotonically with $K$, but $K{=}5$ already achieves strong and stable tracking. Even at $K{=}30$, runtime remains ~8 ms per step—well within real-time constraints. This provides practitioners the flexibility to choose $K$ based on their hardware compute budget. Following **Reviewer 3 (Bd9Z)** 's suggestion, we additionally compared against $L_1$ adaptive control and Tube MPC baselines in the variable-$M$ ablations; our method consistently outperforms all baselines across all configurations, confirming $K$ is not a practical bottleneck.
>
> **PFAR RMS Error (Fixed K=5 , variable M)**
> | **M** | **Wass-MD(ours)**  | **PID** |  **MRR** |  **MLAC** |  **$L_1$ Adaptive** | **Tube MPC** |  **MD**|
> |:---:|:---:|:---:|:---:|:---:|:---:|:---:|:---:|
> | 2  |  **0.1186** | 1.2762 | 1.1980 | 0.5054 | 0.7168 | 1.3677 | 2.1241 |
> | 5  | **0.1242** | 1.2762 | 3.0218 | 0.1821 | 0.3567 | 0.5146 | 0.5405  |
> | 10 |  **0.0859** | 1.2762 | 1.5315 |  0.2334 | 0.4143 | 0.4995 |  0.1826  |
> | 20 | **0.0478** | 1.2762 | 1.8702 | 0.0774 | 0.1998 | 0.3673 | 0.4082  |
> | 30 | **0.1047** | 1.2762 | 1.2144 |  0.1078 | 0.2491 | 0.2628 | 0.6354  |
>
> Tracking trajectories and error plots: [ **PFAR**: https://ibb.co/6Rky12GZ ].
>
> **PVTOL RMS Error (Fixed K=5 , variable M)**
> | **M** | **Wass-MD(ours)** | **PID** | **MRR** |  **MLAC**  |**$L_1$ Adaptive** | **Tube MPC** |
> |:---:|:---:|:---:|:---:|:---:|:---:|:---:|
> | 2  | **0.5724** | 3.0925 | 3.3763 | 1.4177 | 0.8319 | 0.6557 |
> | 5  |  **0.2544** | 3.0925 | 2.7417 | 1.5748 | 0.5962 | 0.5657 |
> | 10 | **0.3834** |  3.0925 | 2.4876 | 1.1854 |  0.6641 | 0.4675 |
> | 20 |  **0.3313** |  3.0925 | 1.9287 | 0.8251 | 0.5761 | 0.4957 |
> | 30 | **0.2852** |  3.0925 | 1.8180 |  0.5430 | 0.5561 | 0.5543 |
>
> Tracking trajectories and error plots: [ **PVTOL**: https://ibb.co/Mybm0b6K ].
>
> **Q3: How should we interpret Figure 5?**
>
> Figure 5 provides a visual representation of how the adaptive parameters of different controllers evolve over time as they learn to compensate for disturbances. Because the actual adaptive parameter matrix $A$ is high-dimensional, PCA is used to project the data into a 3D space. The axes (PC1, PC2, PC3) represent the top three principal component directions, capturing the primary modes of change in parameter evolution. The lines connecting start and end points represent the "search paths" controllers take through parameter space. By observing the tortuousness of these paths, we can directly compare the efficiency of their underlying search geometries.
>
> Traditional parameter searches in Euclidean space frequently struggle against the complex topography of nonlinear dynamics, becoming trapped in inefficient local optima. By lifting adaptation into a non-Euclidean probability space, the Wasserstein framework actively circumvents these geometric obstacles. In contrast to Euclidean paths, the Wass-MD trajectory follows a transport-optimal path in Wasserstein geometry, cutting a distinct, much smoother route through the space. It avoids the wandering loops seen in the baselines, converging far more directly toward its final estimate.
>
> **Limitations and Presentation.** We will include a dedicated Limitations discussion in the additional page allowed during the Camera-Ready stage, incorporating all suggestions raised across reviewers in this rebuttal. We will also improve the visual quality of figures and tables, and add more intuitive physical explanations alongside the mathematical formulations. We sincerely appreciate the reviewer's valuable suggestions on improving the presentation of our paper.

---

> > ### Author Rebuttal · Reviewer_goZj · 2026-04-04
> >
> > Thank you for the detailed response. The K-ablation experiments in Q2 are substantive and convincing, and I appreciate the additional comparisons with L1 Adaptive and Tube MPC.
> >
> > For Q3, I agree with the qualitative interpretation but find it insufficient as evidence. Quantitative metrics such as path length comparisons or convergence rate analysis would strengthen the argument considerably.
> >
> > Regarding presentation, I acknowledge your commitment to improve figures and add intuitive explanations, but these are difficult to evaluate without seeing the actual revisions.
> >
> > I am willing to revise my score modestly in light of the new ablation data, but the concerns above prevent a more significant change at this time.

---

> > > ### Author Response · Authors · 2026-04-06
> > >
> > > We thank you for the detailed follow-up and for acknowledging the K-ablation experiments as "substantive and convincing." Below we address the remaining concern regarding quantitative evidence for Figure 5.
> > >
> > > **Q3 (continued): Quantitative metrics for parameter trajectory analysis**
> > >
> > > Following your suggestion, we have conducted path length comparisons and convergence rate analysis on both PFAR and PVTOL systems. All experiments use the same setup described in Section 6.1 of the main text, ensuring a fair comparison under identical conditions.
> > >
> > > **Path length comparison.** We measure the total arc length of each controller's adaptive parameter trajectory $A(t) \in \mathbb{R}^{m \times d}$ in PCA-projected space. Concretely, we flatten $A(t)$ at each timestep, apply PCA to obtain a 3-dimensional representation $\tilde{A}(t) \in \mathbb{R}^3$, and compute the cumulative Euclidean arc length:
> > > $$\text{PathLength} = \sum_{k=1}^{N-1} \| \tilde{A}(t_{k+1}) - \tilde{A}(t_k) \|_2$$
> > > where $N$ is the number of simulation steps. A shorter path length indicates that the controller reaches its final parameter estimate via a more direct route in parameter space, which is precisely the geometric advantage that Wasserstein transport paths provide over Euclidean updates, as discussed qualitatively in our original response regarding Figure 5.
> > >
> > > On both systems, Wass-MD achieves the shortest parameter path among all methods. The results are summarized below.
> > >
> > > **PFAR Path Length (wind = 8.0 m/s)**
> > >
> > > | **Wass-MD(ours)**  | **PID** |  **MRR** |  **MLAC** |  **$L_1$ Adaptive** | **Tube MPC** |  **MD**|
> > > |:---:|:---:|:---:|:---:|:---:|:---:|:---:|
> > > | **24.6** | 119.9 | 34.1 | 47.8 | 34.3 | 53.5 | 50.6 |
> > >
> > > On PFAR, Wass-MD  reduces the path length by 28% compared to the next best adaptive method MRR  and by 79% compared to PID.
> > >
> > > **PVTOL Path Length (peak wind = 8.0 m/s)**
> > >
> > > | **Wass-MD(ours)** | **PID** | **MRR** |  **MLAC**  |**$L_1$ Adaptive** | **Tube MPC** |
> > > |:---:|:---:|:---:|:---:|:---:|:---:|
> > > |  **7.2** | 24.3 | 16.4 | 9.3 | 15.3 | 21.1 |
> > >
> > >  On PVTOL, Wass-MD  is 23% shorter than MLAC. These results confirm that the Wasserstein geometry guides parameter updates along transport-optimal paths, reaching effective parameter estimates through substantially more direct trajectories than Euclidean or static p-norm alternatives.
> > >
> > > We additionally provide cumulative path length plots showing how each method's trajectory accumulates over the full simulation horizon. These plots reveal that Wass-MD maintains a consistently lower accumulation rate throughout the control period, indicating sustained geometric efficiency. The path length analysis for both systems is available at:
> > >
> > > - PFAR path length analysis: https://ibb.co/mrY9JZZV
> > > - PVTOL path length analysis: https://ibb.co/HLRpcBhX
> > >
> > > **Convergence rate analysis.** We provide tracking error convergence plots $\|e(t)\|_2$ for both systems under identical conditions. On PFAR ($w=8.0$ m/s, $T=10$ s), Wass-MD maintains the lowest tracking error throughout, while MRR peaks above 2.5 and PID oscillates around 1.0-1.7. On PVTOL (time-varying wind peaking at 8.0 m/s, $T=20$ s), all methods experience increasing errors during the high-wind phase, but Wass-MD shows the smallest magnitude and fastest recovery, compared to peaks of 5.8 (PID) and 4.5 (MRR). These convergence curves are available at:
> > >
> > > - PFAR convergence analysis: https://ibb.co/77MvZnC
> > > - PVTOL convergence analysis: https://ibb.co/8nHr1GTz
> > >
> > > Together, the path length comparison and convergence analysis provide the quantitative evidence : the shorter parameter paths of Wass-MD directly correspond to faster and more stable tracking error convergence, confirming that the qualitative observation from Figure 5 (smoother, more direct parameter trajectories) translates into measurable performance gains.
> > >
> > > **Regarding presentation improvements.** In the Camera-Ready revision, we will: (1) redraw all experimental figures with aspect ratios to eliminate the stretched-font issue; (2) add physical intuition paragraphs before major theorems (e.g., explaining the role of kernel bandwidth $\sigma$ in balancing exploration and exploitation among particles before presenting the stability bound); (3) include a dedicated Limitations section incorporating all reviewers' suggestions.
> > >
> > > We hope that these quantitative analyses, together with the K-ablation and comparisons results from our previous response, provide clear and direct evidence supporting our method's geometric advantages. We believe these improvements significantly strengthen both the presentation and the empirical evidence of our work. We would be grateful if you could consider reflecting this progress by raising the score.
> > >
> > > Thank you again for your constructive feedback, which has been instrumental in improving our manuscript. Should you have any further thoughts or suggestions during the remaining discussion period, we are happy to address them.
> > >
> > > Best regards,
> > >
> > > Authors

---

### Decision · Program_Chairs · 2026-04-30

**Decision:**

Accept (regular)

**Comment:**

This paper proposes a framework for adaptive control of nonlinear systems that lifts the adaptation law from finite-dimensional parameter spaces into Wasserstein space, constructing a meta-learned Wasserstein Bregman divergence over representative task distributions and applying the resulting controller to both fully-actuated and underactuated planar rotorcraft under significant distributional shift between training and test conditions.

Reviewers broadly recognized the novelty of the approach, the quality of the empirical evaluation, and the value of extending meta-learned mirror descent adaptive control to the underactuated setting, which prior work had not addressed.

Reviewer Bd9Z raises a substantive concern that the paper's framing is stronger than what the proofs support.  Specifically, while the Lyapunov candidate is formulated in terms of the Wasserstein Bregman divergence induced by the occupancy measure \mu, the stability certificate as proved reduces for the empirical and Dirac measures actually used to an average of standard Bregman divergences in which \mu does not appear, with the uniformly ultimately bounded guarantee depending on the kernel bandwidth \sigma rather than on any \mu-dependent geometric quantity. The authors should further note, as Bd9Z observes, that the K=1, \sigma \to 0 special case recovers the predecessor method with strictly stronger asymptotic tracking guarantees rather than uniformly ultimately bounded ones, a fact acknowledged in the appendix but not adequately discussed in the paper's characterization of its contributions relative to prior work.

Conditional on the authors resolving this gap, either by tightening the theoretical narrative to accurately characterize where \mu enters the framework or by strengthening the analysis so that the Wasserstein geometry plays a more direct role in the stability certificate, the paper makes a meaningful contribution to the control-oriented meta-learning literature and is recommended for acceptance.